# Modulated Neural ODEs

**Ilze Amanda Auzina**
University of Amsterdam
i.a.auzina@uva.nl

**Çağatay Yıldız**
University of Tübingen
Tübingen AI Center

**Sara Magliacane**
University of Amsterdam
MIT-IBM Watson AI Lab

**Matthias Bethge**
University of Tübingen
Tübingen AI Center

**Efstratios Gavves**
University of Amsterdam

## Abstract

Neural ordinary differential equations (NODEs) have been proven useful for learning non-linear dynamics of arbitrary trajectories. However, current NODE methods capture variations across trajectories only via the initial state value or by auto-regressive encoder updates. In this work, we introduce Modulated Neural ODEs (MoNODEs), a novel framework that sets apart dynamics states from underlying static factors of variation and improves the existing NODE methods. In particular, we introduce *time-invariant modulator variables* that are learned from the data. We incorporate our proposed framework into four existing NODE variants. We test MoNODE on oscillating systems, videos and human walking trajectories, where each trajectory has trajectory-specific modulation. Our framework consistently improves the existing model ability to generalize to new dynamic parameterizations and to perform far-horizon forecasting. In addition, we verify that the proposed modulator variables are informative of the true unknown factors of variation as measured by $R^2$ scores.

## 1 Introduction

Differential equations are the *de facto* standard for learning dynamics of biological [Hirsch et al., 2012] and physical [Tenenbaum and Pollard, 1985] systems. When the observed phenomenon is deterministic, the dynamics are typically expressed in terms of ordinary differential equations (ODEs). Traditionally, ODEs have been built from a mechanistic perspective, in which states of the observed system and the governing differential equation with its parameters are specified by domain experts. However, when the parametric form is unknown and only observations are available, neural network surrogates can be used to model the unknown differential function, called neural ODEs (NODEs) Chen et al. [2018]. Since its introduction by Chen et al. [2018], there has been an abundance of research that uses NODE type models for learning differential equation systems and extend it by introducing a recurrent neural networks (RNNs) encoder [Rubanova et al., 2019, Kanaa et al., 2021], a gated recurrent unit [De Brouwer et al., 2019, Park et al., 2021], or a second order system [Yildiz et al., 2019, Norcliffe et al., 2020, Xia et al., 2021].

Despite these recent developments, all of the above methods have a common limitation: any static differences in the observations can only be captured in a time-evolving ODE state. This modelling approach is not suitable when the observations contain underlying factors of variation that are fixed in time, yet can (i) affect the dynamics or (ii) affect the appearance of the observations. As a concrete example of this, consider human walking trajectories. The overall motion is shared across all subjects, e.g. walking. However, every subject might exhibit person-specific factors of variation, which in turn could either affect the motion exhibited, e.g. length of legs, or could be a characteristic of a given subject, e.g. color of a shirt. A modelling approach that is able to (i) distinguish the dynamic

37th Conference on Neural Information Processing Systems (NeurIPS 2023).

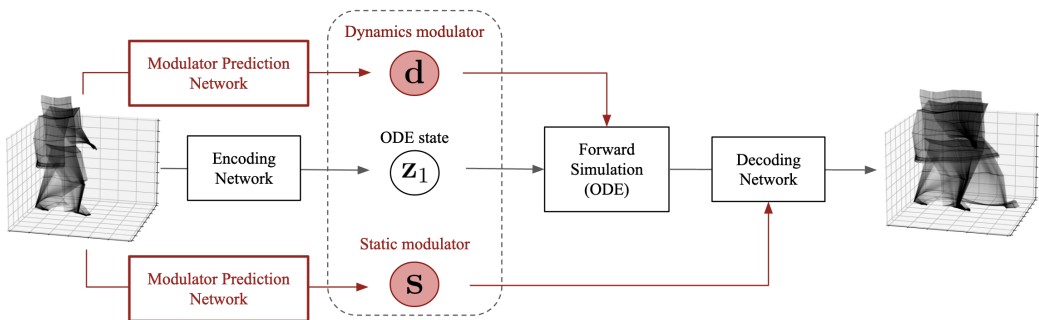

Figure 1: Schematic illustration of our method. Standard continuous-time latent dynamical systems such as Chen et al. [2018] assume a single ODE state $\mathbf{z}(t)$ transformed by a surrogate neural differential function. Our method augments the latent space with dynamics and static modulators to account for exogenous variables governing the dynamics and observation mapping (decoding).

(e.g. position, velocity) from the static variables (e.g. height, clothing color), and (ii) use the static variables to modulate the motion or appearance, is advantageous, because it leads to an improved generalization across people. As an empirical confirmation, we show experimentally that the existing NODE models fail to separate the dynamic factors from the static factors. This inevitably leads to overfitting, thus, negatively effecting the model generalization to new dynamics, as well as far-horizon forecasting.

As a remedy, this work introduces a modulated neural ODE (MONODE) framework, which separates the dynamic variables from the time-invariant variables. We call these time-invariant variables *modulator variables* and we distinguish between two types: *(i) static modulators* that modulate the appearance; and *(ii) dynamics modulators* that modulate the time-evolution of the latent dynamical system (for a schematic overview see Fig. 1). In particular, MONODE adds a *modulator prediction network* on top of a NODE, which allows to compute the *modulator variables* from data. We empirically confirm that our modular framework boosts existing NODE models by achieving improved future predictions and improved generalization to new dynamics. In addition, we verify that our modulator variables are more informative of the true unknown factors of variation by obtaining higher $R^2$ scores than NODE. As a result, the latent ODE state of MONODE has an equivalent sequence-to-sequence correspondence as the true observations. Our contributions are as follows:

- We extend neural ODE models by introducing *modulator variables* that allow the model to preserve core dynamics while being adaptive to modulating factors.
- Our modulator framework can easily be integrated into existing NODE models [Chen et al., 2018, Yildiz et al., 2019, Norcliffe et al., 2020, Xia et al., 2021].
- As we show in our experiments on sinusoidal waves, predator-prey dynamics, bouncing ball videos, rotating images, and motion capture data, our modulator framework consistently leads to improved long-term predictions as measured by lower test mean squared error (an average of 55.25% improvement across all experiments).
- Lastly, we verify that our modulator variables are more informative of the true unknown factors of variation than NODE, as we show in terms of $R^2$ scores.

While introduced in the context of neural ODEs, we believe that the presented framework can benefit also stochastic and/or discrete dynamical systems, as the concept of dynamic or mapping modulation is innate for most dynamical systems. We conclude this work by discussing the implications of our method on learning object-centric representations and its potential Bayesian extensions. Our official implementation can be found at `https://github.com/IlzeAmandaA/MoNODE`.

## 2 Background

We first introduce the basic concepts for Neural ODEs, following the notation by Chen et al. [2018].

**Ordinary differential equations**    Multivariate ordinary differential equations are defined as

$$\dot{\mathbf{z}}(t) := \frac{\mathrm{d}\mathbf{z}(t)}{\mathrm{d}t} = \mathbf{f}(t, \mathbf{z}(t)), \tag{1}$$

where $t \in \mathbb{R}_+$ denotes *time*, the vector $\mathbf{z}(t) \in \mathbb{R}^{q_z}$ captures the *state* of the system at time $t$, and $\dot{\mathbf{z}}(t) \in \mathbb{R}^{q_z}$ is the *time derivative* of the state $\mathbf{z}(t)$. In this work, we focus on autonomous ODE systems, implying a vector-valued *(time) differential function* $\mathbf{f} : \mathbb{R}^{q_z} \mapsto \mathbb{R}^{q_z}$ that does not explicitly depend on time. The ODE state solution $\mathbf{z}(t_2)$ is computed by integrating the differential function starting from an initial value $\mathbf{z}(t_1)$:

$$\mathbf{z}(t_2) = \mathbf{z}(t_1) + \int_{t_1}^{t_2} \mathbf{f}(\mathbf{z}(\tau))\mathrm{d}\tau. \tag{2}$$

For non-linear differential functions $f$, the integral does not have a closed form solution and hence is approximated by numerical solvers [Tenenbaum and Pollard, 1985]. Due to the deterministic nature of the differential function, the ODE state solution $\mathbf{z}(t_2)$ is completely determined by the corresponding initial value $\mathbf{z}(t_1)$ if the function $\mathbf{f}$ is known.

## 2.1 Latent neural ODEs

Chen et al. [2018] proposed neural ODEs for modeling sequential data $\mathbf{x}_{1:T} \in \mathbb{R}^{T \times D}$, where $\mathbf{x}_i \equiv \mathbf{x}(t_i)$ is the $D$-dimensional observation at time $t_i$, and $T$ is the sequence length. We assume known observation times $\{t_1, \ldots, t_T\}$. Being a latent variable model, NODE infers a *latent trajectory* $\mathbf{z}_{1:T} \in \mathbb{R}^{T \times q_z}$ for an input trajectory $\mathbf{x}_{1:T}$. The generative model relies on random initial values, their continuous-time transformations, and finally an observation mapping from latent to data space:

$$\mathbf{z}_1 \sim p(\mathbf{z}_1) \tag{3}$$

$$\mathbf{z}_i = \mathbf{z}_1 + \int_{t_1}^{t_i} \mathbf{f}_{\boldsymbol{\theta}}(\mathbf{z}(\tau))\mathrm{d}\tau \tag{4}$$

$$\mathbf{x}_i \sim p_{\boldsymbol{\xi}}(\mathbf{x}_i \mid \mathbf{z}_i) \tag{5}$$

Here, the time differential $\mathbf{f}_{\boldsymbol{\theta}}$ is a neural network with parameters $\boldsymbol{\theta}$ (hence the name "neural ODEs"). Similar to variational auto-encoders [Kingma and Welling, 2013, Rezende et al., 2014], the "decoding" of the observations is performed by another non-linear neural network with a suitable architecture and parameters $\boldsymbol{\xi}$.

# 3 MONODE: Modulated Neural ODEs

We begin with a brief description of the dynamical systems of our interest. Without loss of generality, we consider a dataset of $N$ trajectories with a fixed trajectory length $T$, where the $n$'th trajectory is denoted by $\mathbf{x}_{1:T}^n$. First, we make the common assumption that the data trajectories are generated by a single dynamical system (*e.g.*, a swinging pendulum) while the parameters that *modulate the dynamics* (*e.g.*, pendulum length) vary across the trajectories. Second, we focus on the more general setup in which the observations and the dynamics might lie in different spaces, for example, video recordings of a pendulum. A future video prediction would require a mapping from the dynamics space to the observation space. As the recordings might exhibit different lighting conditions or backgrounds, the mapping typically involves static features that *modulate the mappings*, *e.g.*, a parameter specifying the background.

## 3.1 Our generative model

The generative model of NODE [Chen et al., 2018] involves a latent initial value $\mathbf{z}_1$ for each observed trajectory as well as a differential function and a decoder with global parameters $\boldsymbol{\theta}$ and $\boldsymbol{\xi}$. Hence, by construction, NODE can attribute discrepancies across trajectories only to the initial value as the remaining functions are modeled globally. Subsequent works [Rubanova et al., 2019, Dupont et al., 2019, De Brouwer et al., 2019, Xia et al., 2021, Iakovlev et al., 2023] combine the latent space of NODE with additional variables, however, likewise, they do not account for static, trajectory-specific factors that modulate either the dynamics or the observation mapping. The main claim of this work is that explicitly modeling the above-mentioned *modulating* variables results in better extrapolation and

generalization abilities. To show this, we introduce the following generative model:

$$\mathbf{d} \sim p(\mathbf{d}) \quad \textit{// dynamics modulator} \tag{6}$$

$$\mathbf{s} \sim p(\mathbf{s}) \quad \textit{// static modulator} \tag{7}$$

$$\mathbf{z}_1 \sim p(\mathbf{z}_1) \quad \textit{// latent ODE state} \tag{8}$$

$$\mathbf{z}_i = \mathbf{z}_1 + \int_{t_1}^{t_i} \mathbf{f}_{\boldsymbol{\theta}}(\mathbf{z}(\tau); \mathbf{d}) \, \mathrm{d}\tau \tag{9}$$

$$\mathbf{x}_i \sim p_{\boldsymbol{\xi}}(\mathbf{x}_i \mid \mathbf{z}_i \, ; \, \mathbf{s}). \tag{10}$$

We broadly refer to $\mathbf{d}$ and $\mathbf{s}$ as *dynamics* and *static modulators*, and thus name our framework *Modulated Neural ODE* (MONODE). We assume that each observed sequence $\mathbf{x}_{1:T}^n$ has its own modulators $\mathbf{d}^n$ and $\mathbf{s}^n$. As opposed to the ODE state $\mathbf{z}(t)$, the modulators are *time-invariant*.

We note that for simplicity, we describe our framework in the context of the initial neural ODE model [Chen et al., 2018]. However, our framework can be readily adapted to other neural ODE models such as second-order and heavy ball, NODEs, as we demonstrate experimentally. For a schematic overview, please see Fig. 1. Next, we discuss how to learn the *modulator variables* along with the global dynamics, encoder, and decoder.

## 3.2   Learning latent modulator variables

A straightforward approach to obtain time-invariant *modulator variables* is to define them globally and independently of each other and input sequences. While optimizing for such global variables works well in practice [Blei et al., 2017], it does not specify how to compute variables for an unobserved trajectory. As the focus of the present work is on improved generalization to unseen dynamic parametrizations, we estimate the *modulator variables* via amortized inference based on encoder networks $\mathbf{g}_{\boldsymbol{\psi}}$ and $\mathbf{h}_{\boldsymbol{\upsilon}}$, which we detail in the following.

**(i) Static modulator**   To learn the unknown static modulator $\mathbf{s}^n \in \mathbb{R}^{q_s}$ that captures the time-invariant characteristics of the individual observations $\mathbf{x}_i^n$, we compute the average over the observation embeddings provided by a modulator prediction network $\mathbf{g}_{\boldsymbol{\psi}}(\cdot)$ (*e.g.*, a convolutional neural network) with parameters $\boldsymbol{\psi}$:

$$\mathbf{s}^n = \frac{1}{T} \sum_{i=1}^{T} \mathbf{s}_i^n, \qquad \mathbf{s}_i^n = \mathbf{g}_{\boldsymbol{\psi}}(\mathbf{x}_i^n). \tag{11}$$

By construction, $\mathbf{s}^n$ is time-invariant (or more rigorously, invariant to time-dependent effects) as we average over time. In turn, the decoder takes as input the concatenation of the latent ODE state $\mathbf{z}_i^n$ and the static modulator $\mathbf{s}^n$, and maps the joint latent representation to the observation space (similarly to Franceschi et al. [2020]):

$$\mathbf{x}_i^n \sim p_{\boldsymbol{\xi}}\left(\mathbf{x}_i^n \Big| [\mathbf{z}_i^n, \mathbf{s}^n]\right) \ \ \forall i \in [1, \dots, T]. \tag{12}$$

Note that the estimated static modulator $\mathbf{s}^n$ is fed as input for all time points within a trajectory.

**(ii) Dynamics modulator**   Unlike the static modulator, the dynamics modulator can only be deduced from multiple time points $\mathbf{x}_{i+T_e}^n$. For example, the dynamics of a pendulum depend on its length. To compute the length of the pendulum one must compute the acceleration for which multiple position and velocity measurements are needed. Thereby, the dynamics modulators, $\mathbf{d}_i$, are computed from subsequences of length $T_e$ from a given trajectory. To achieve time-invariance we likewise average over time:

$$\mathbf{d}^n = \frac{1}{T - T_e} \sum_{i=1}^{T-T_e} \mathbf{d}_i^n, \qquad \mathbf{d}_i^n = \mathbf{h}_{\boldsymbol{\upsilon}}(\mathbf{x}_{i:i+T_e}^n), \tag{13}$$

where $\mathbf{h}_{\boldsymbol{\upsilon}}$ is a modulator prediction network (*e.g.*, a recurrent neural network) with parameters $\boldsymbol{\upsilon}$. The differential function takes as input the concatenation of the latent ODE state $\mathbf{z}^n(\tau) \in \mathbb{R}^{q_z}$ and the estimated dynamics modulator $\mathbf{d}^n \in \mathbb{R}^{q_d}$. Consequently, we redefine the input space of the differential function $\mathbf{f}_{\boldsymbol{\theta}} : \mathbb{R}^{q_z + q_d} \mapsto \mathbb{R}^{q_z}$, implying the following time differential:

$$\frac{\mathrm{d}\mathbf{z}^n(t)}{\mathrm{d}t} = \mathbf{f}_{\boldsymbol{\theta}}(\mathbf{z}^n(t), \mathbf{d}^n) \tag{14}$$

Table 1: A taxonomy of the state-of-the-art ODE systems. We empirically demonstrate that our modulating variables idea is straightforward to apply to NODE, SONODE, and HBNODE, and consistently leads to improved prediction accuracy. Where with * we refer to: MONODE, MOSONODE, MOLSONODE, MOHBNODE.

| Model | Reference | Latent dynamics | Time invariant modulator | Temporal data |
|---|---|---|---|---|
| Neural ODE (NODE) | Chen et al. [2018] | ✓ | ✗ | ✓ |
| Augmented NODE (ANODE) | Dupont et al. [2019] | ✗ | ✗ | ✗ |
| Neural Controlled DE (NCDE) | Kidger et al. [2020] | ✓ | ✗ | ✗ |
| Second Order NODE (SONODE) | Norcliffe et al. [2020] | ✗ | ✗ | ✓ |
| Latent SONODE (LSONODE) | Yildiz et al. [2019] | ✓ | ✗ | ✓ |
| NODE Processes (NODEP) | Norcliffe et al. [2021] | ✓ | ✗ | ✓ |
| Heavy Ball NODE (HBNODE) | Xia et al. [2021] | ✗ | ✗ | ✓ |
| Modulated *NODE | this work | ✓ | ✓ | ✓ |

The resulting ODE system resembles in a way the augmented neural ODE (ANODE) [Dupont et al., 2019]. However, their appended variable dimensions are constant and serve a practical purpose of breaking down the diffeomorphism constraints of NODE, while ours models time-invariant variables. We treat $T_e$ as a hyperparameter and choose it by cross-validation.

**Optimization objective** The maximization objective of MONODE is analogous to the evidence-lower bound (ELBO) as in [Chen et al., 2018] for NODE, where we place a prior distribution on the unknown latent initial value $\mathbf{z}_1$ and approximate it by amortized inference. Similar to previous works [Chen et al., 2018, Yildiz et al., 2019], MONODE encoder for $\mathbf{z}_1$ takes a sequence of length $T_z < T$ as input, where $T_z$ is a hyper-parameter. We empirically observe that our framework is not sensitive to $T_z$. The optimization of the modulator prediction networks $\mathbf{g}_\psi$ and $\mathbf{h}_\upsilon$ is implicit in that they are trained jointly with other modules while maximizing the ELBO objective.

## 4 Related work

**Neural ODEs** Since the neural ODE breakthrough [Chen et al., 2018], there has been a growing interest in continuous-time dynamic modeling. Such attempts include combining recurrent neural nets with neural ODE dynamics [Rubanova et al., 2019, De Brouwer et al., 2019], where latent trajectories are updated upon observations, as well as upon Hamiltonian [Zhong et al., 2019], Lagrangian [Lutter et al., 2019], second-order [Yildiz et al., 2019], or graph neural network based dynamics [Poli et al., 2019]. While our method MONODE has been introduced in the context of latent neural ODEs, it can be directly utilized within these frameworks as well.

**Augmented dynamics** Dupont et al. [2019] augment data-space neural ODEs with additional latent variables and test their method on classification problems. Norcliffe et al. [2021] extend neural ODEs to stochastic processes by means of stochastic latent variables, leading to NODE Processes (NODEP). By construction, NODEP embeddings are invariant to the shuffling of the observations. To the best of our knowledge, we are the first to explicitly enforce *time-invariant modulator variables*.

**Learning time-invariant variables** The idea of averaging for invariant function estimation was used in [Kondor, 2008, van der Wilk et al., 2018, Franceschi et al., 2020]. Only the latter proposes using such variables in the context of discrete-time stochastic video prediction. Although relevant, their model involves two sets of dynamic latent variables, coupled with an LSTM and is limited to mapping modulation.

## 5 Experiments

To investigate the effect of our proposed *dynamics* and *static modulators*, we structure the experiments as follows: First, we investigate the effect of the *dynamics modulator* on classical dynamical systems, namely, sinusoidal wave, predator-prey trajectories and bouncing ball (section 5.1), where the

Table 2: Test MSE and its standard deviation with and without our framework for NODE [Chen et al., 2018], SONODE [Norcliffe et al., 2020], and HBNODE Xia et al. [2021]. We test model performance within the training regime and for forecasting accuracy. Each model is run three times with different initial seed, we report the mean and standard deviation across runs. Lower is better.

| Model | Sinusoidal data | | Predator-prey data | |
|---|---|---|---|---|
| | $T = 50$ | $T = 150$ | $T = 100$ | $T = 300$ |
| NODE | 0.13 (0.03) | 1.84 (0.70) | 0.85 (0.08) | 23.81 (2.29) |
| MoNODE (ours) | **0.04** (0.01) | **0.29** (0.11) | **0.74** (0.05) | **4.33** (0.19) |
| SONODE | 2.19 ( 0.15) | 3.05 ( 0.07) | 15.80 (0.75) | 40.92 (0.82) |
| MoSONODE (ours) | **0.05** ( 0.01) | **0.35** ( 0.10) | **1.46** (0.28) | **6.70** (0.83) |
| HBNODE | 0.16 ( 0.02) | 3.36 ( 0.33) | 0.88 ( 0.10) | 3346.62 (2119.24) |
| MoHBNODE (ours) | **0.05** ( 0.01) | **0.65** ( 0.30) | 0.94 ( 0.09) | **10.21** ( 1.43) |

parameterisation of the dynamics differs across each trajectory. Second, to confirm the utility of the *static modulator* we implement an experiment of rotating MNIST digits (section 5.2), where the static content is the digit itself. Lastly, we experiment on real data with having both modulator variables present for predicting human walking trajectories (section 5.4). In all experiments, we test whether our framework improves the performance of the base model on generalization to new trajectories and long-horizon forecasting abilities.

**Implementation details**   We implement all models in PyTorch [Paszke et al., 2017]. The encoder, decoder, differential function, and modulator prediction networks are all jointly optimized with the Adam optimizer [Kingma and Ba, 2014]. For solving the ODE system we use `torchdiffeq` [Chen, 2018] package. We use the 4th-order Runge-Kutta numerical solver to compute ODE state solutions (see App. D for ablation results for different solvers). For the complete details on data generation and training setup, we refer to App.B. Further, we report the architectures, number of parameters, and details on hyperparameter for each method in App. C Table 7.

**Compared methods**   We test our framework on the following models: (i) Latent neural ODE model [Chen et al., 2018] (NODE), (ii) Second-order NODE model [Norcliffe et al., 2020] (SONODE), (iii) Latent second-order NODE model [Yildiz et al., 2019] (LSONODE), (iv) current state-of-the-art, second-order heavy ball NODE [Xia et al., 2021] (HBNODE).

In order for SONODE and HBNODE to have a comparable performance with NODE we adjust the original implementation by the authors by changing the encoder architecture, while keeping the core of the models, the differential function, unchanged. For further details and discussion, see App. A. We do not compare against ANODE [Dupont et al., 2019] as the methodology is presented in the context of density estimation and classification. Furthermore, we performed preliminary tests with NODEP [Norcliffe et al., 2021]; however, the model predictions fall back to the prior in datasets with dynamics modulators. Hence, we did not include any results with this model in the paper as the base model did not have sufficiently good performance. Finally, we chose not to compare against Kidger et al. [2020] as their Riemann–Stieltjes integral relies on smooth interpolations between data points while we particularly focus on the extrapolation performance for which the ground truth data is not available. For an overview of the methods discussed see Table 1.

## 5.1 Dynamics modulator variable

To investigate the benefit of the *dynamics modulator* variable, we test our framework on three dynamical systems: sine wave, prey-predator (PP) trajectories, and bouncing ball (BB) videos. In contrast to earlier works [Norcliffe et al., 2020, Rubanova et al., 2019], every trajectory has a **different parameterization** of the differential equation. Intuitively, the goal of the modulator prediction network is to learn this parameterisation and pass it as an input to the dynamics function, which is modelled by a neural network.

### 5.1.1 Sinusoidal data

The training data consists of $N = 300$ oscillating trajectories with length $T = 50$. The amplitude of each trajectory is sampled as $a^n \sim \mathbb{U}[1, 3]$ and the frequency is sampled as $\omega^n \sim \mathbb{U}[0.5, 1.0]$,

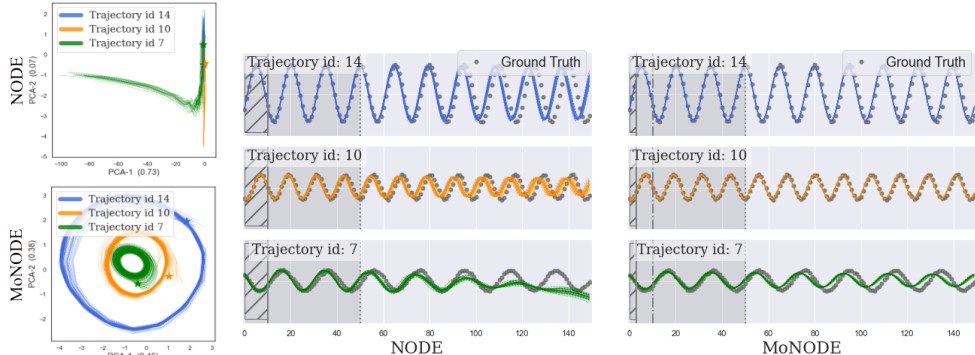

Figure 2: The left panel illustrates PCA embeddings of the latent ODE state inferred by NODE and MONODE, where each color matches one trajectory. The figures on the right compare NODE and MONODE reconstructions on test trajectory sequences. Note that the models take the dashed area as input, and the darker area represents the trajectory length seen during training.

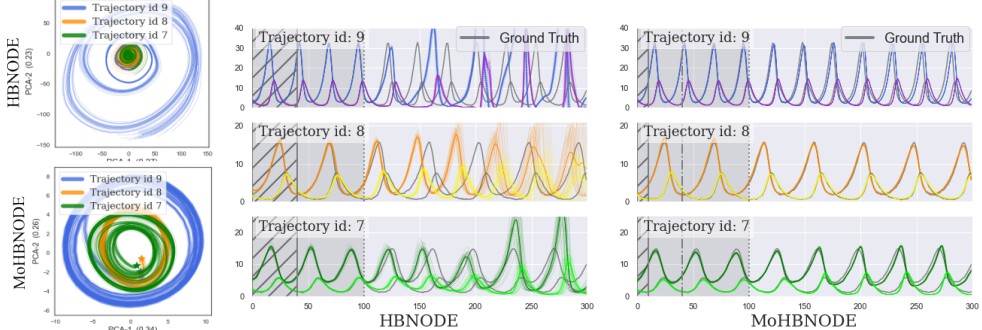

Figure 3: To show our framework applicability to different NODE variants, here we illustrate HBNODE and MOHBNODE performance on PP data. The left panel illustrates PCA embeddings of the latent ODE state inferred by HBNODE and MOHBNODE, where each color matches one trajectory. The figures on right compare HBNODE and MOHBNODE reconstructions on test trajectory sequences. Note that the models take the dashed area as input, and darker area represents the trajectory length seen during training. predator-prey data.

where $\mathbb{U}[\cdot, \cdot]$ denotes a uniform distribution. Validation and test data consist of $N_{\text{val}} = N_{\text{test}} = 50$ trajectories with sequence length $T_{\text{val}} = 50$ and $T_{\text{test}} = 150$, respectively. We add noise to the data following the implementation of [Rubanova et al., 2019].

The obtained test MSE demonstrates that our framework improves all aforementioned methods, namely, NODE, SONODE, and the state-of-the-art HBNODE (see Fig. 2, Fig. 11, and Fig. 12). In particular, our framework improves generalization to new dynamics and far-horizon forecasting as reported in Table 2 columns 2 and 3. In addition, modulating the motion via the dynamics modulator variable leads to interpretable latent ODE state trajectories $\mathbf{z}(t)$, see Fig. 2. More specifically, the obtained latent trajectories $\mathbf{z}(t)$ have qualitatively a comparable radius topology to the observed amplitudes in the data space. By contrast, the latent space of NODE does not have such structure. Similar results are obtained for MOHBNODE (App. D, Fig. 12).

**Training and inference times**    To showcase that our proposed framework is easy to train we have plotted the validation MSE versus wall clock time during training for sinusoidal data, please see App. Fig. 10. As it is apparent from the figure, our framework is easier to train than all baseline methods. We further compute the inference time cost for the sin data experiment, where the test data consists of 50 trajectories of length 150. We record the time it takes NODE and MONODE to predict future states while conditioned on the initial 10 time points. Repeating the experiment ten times, the inference time cost for NODE is $0.312 \pm 0.050$ while for MONODE is $0.291 \pm 0.040$.

### 5.1.2    Predator-prey (PP) data

Next, we test our framework on the predator-prey benchmark [Rubanova et al., 2019, Norcliffe et al., 2021], governed by a pair of first-order nonlinear differential equations (also known as Lotka-Volterra

system, Eq. 28). The training data consists of $N = 600$ trajectories of length $T = 100$. For every trajectory the four parameters of the differential equation are sampled, therefore each trajectory specifies a different interaction between the two populations. Validation and test data consist of $N_{\text{val}} = N_{\text{test}} = 100$ trajectories with sequence length $T_{\text{val}} = 100$ and $T_{\text{test}} = 300$. Similarly to sinusoidal data, we add noise to the data following [Rubanova et al., 2019].

The results show that our modulator framework improves the test accuracy of the existing methods for both, generalization to new dynamics as well as for forecasting, see Table 2. Moreover, examining the latent ODE state embeddings reveals that our framework results in more interpretable latent space embeddings also for PP data, see Fig. 3 for HBNODE and App. D Fig. 13 for NODE. In particular, the latent space of MOHBNODE and MONODE captured the same amplitude relationship across trajectories as in observation space. For visualisation of the SONODE, MOSONODE, NODE, and MONODE trajectories see App. D, Fig. 14, Fig. 13.

### 5.1.3 Bouncing ball (BB) with friction

To investigate the performance of our framework on video sequences we test it on a bouncing ball dataset, a benchmark often used in temporal generative modeling [Sutskever et al., 2008, Gan et al., 2015, Yildiz et al., 2019]. For data generation, we modify the original implementation of Sutskever et al. [2008] by adding friction to every data trajectory, where friction is sampled from $\mathbb{U}[0, 0.1]$. The friction slows down the ball by a constant factor and is to be inferred by the dynamics modulator. We use $N = 1000$ training sequences of length $T = 20$, and $N_{\text{val}} = N_{\text{test}} = 100$ validation and test trajectories with length $T_{\text{val}} = 20$ and $T_{\text{test}} = 40$. Our framework improves predictive capability for video sequences as shown in Table 4. As visible in Fig .15, the standard NODE model fails to predict the position of the object at further time points, while MONODE corrects this error. For the second-order variants, LSONODE and MOLSONODE, we again observe our framework improving MSE from $0.0181$ to $0.014$.

### 5.1.4 Informativeness metric

Next, we quantify how much information latent representations carry about the unknown factors of variation (FoVs) [Eastwood and Williams, 2018], which are the parameters of the sinusoidal, PP, and BB dynamics. As described in [Schott et al., 2021], we compute $R^2$ scores by regressing from latent variables to FoVs. The regression inputs for MONODE are the dynamics modulators $\mathbf{d}$, while for NODE the latent trajectories $\mathbf{z}_{1:T}$. Note that $R^2 = 1$ corresponds to perfect regression and $R^2 = 0$ indicates random guessing. Our framework ob-

Table 3: $R^2$ scores to predict the unknown FoVs from inferred latents. Higher is better.

|  | NODE | MONODE |
|---|---|---|
| Sine | 0.90 | 0.99 |
| PP | -1.35 | 0.39 |
| BB | -0.29 | 0.58 |

tains better $R^2$ scores on all benchmarks (Table 3), implying better generalization capability of our framework. Therefore, as stated in [Eastwood and Williams, 2018], our MONODE is better capable of disentangling underlying factors of variations compared to NODE.

### 5.2 Static modulator variable

To investigate the benefit of the *static modulator* variable, we test our framework on Rotating MNIST dataset, where the dynamics are fixed and shared across all trajectories, however, the content varies. The goal of the modulator prediction network is to learn the static features of each trajectory and pass it as an input to the decoder network. The data is generated following the implementation by [Casale et al., 2018], where the total number of rotation angles is $T = 16$. We include all ten digits and the initial rotation angle is sampled from all possible angles $\theta^n \sim \{1, \ldots, 16\}$. The training data consists of $N = 1000$ trajectories with length $T = 16$, which corresponds to one cycle of rotation. Validation and test data consist of $N_{\text{val}} = N_{\text{test}} = 100$ trajectories with sequence length $T_{\text{val}} = T_{\text{test}} = 45$. At test time, the model receives the first $T_z$ time frames as input and predicts the full horizon ($T = 45$). We repeat each experiment 3 times and report the mean and standard deviation of the MSEs computed on the forecasting horizon (from $T = 15$ to $T = 45$). For further training details see App.B, while for a complete overview of hyperparameter see App. C Table 7.

The obtained test MSE confirms that the *static modulator* variable improves forecasting quality ($T = 45$), see Table 4 and App. D fig. 16 for qualitative comparison. In addition, the latent ODE

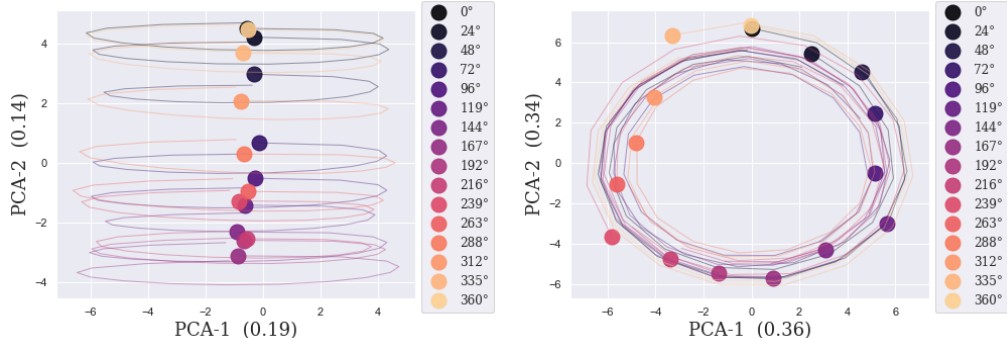

Figure 4: PCA embeddings of the latent ODE state for Rotating MNIST dataset as inferred by NODE (top) and MONODE (bottom). We generate 16 trajectories from a single digit where the initial angle of every trajectory is incremented by $24°$ degrees, starting from $0°$ until $360°$ degrees. Circle: the start of the trajectory (the initial angle), line: ODE trajectory. The color gradient corresponds to the initial angle of the trajectory in observation space.

Table 4: A Comparison of NODE and MONODE test MSEs and standard deviations on BOUNCING BALL, ROT.MNIST, and MOCAP datasets.

|  | BOUNCING BALL | ROT.MNIST | MOCAP | MOCAP-SHIFT |
|---|---|---|---|---|
| NODE | 0.0199(0.001) | 0.039 (0.003) | 72.2(12.4) | 61.6(6.2) |
| MONODE | 0.0164(0.001) | 0.030 (0.001) | 57.7(9.8) | 58.0(10.7) |

states of MONODE form a circular rotation pattern resembling the observed dynamics in data space while for NODE no correspondence is observed, see App. D Fig. 16. Moreover, as shown in Fig. 4, the latent space of MONODE captured the relative distances between the initial rotation angles, while NODE did not. The TSNE embeddings of the *static modulator* indicate a clustering per digit shape, see App. D Fig. 17. For additional results and discussion, see App. D.

### 5.3 Are modulators interchangeable?

To confirm the role of each modulator variable we have performed two additional ablations with the MONODE framework on: *(a)* sinusoidal data with static modulator instead of dynamics, and *(b)* rotating MNIST with dynamics modulator instead of static. We report the test MSE across three different initialization runs with standard deviation. For sinusoidal data with MONODE + static modulator, the test MSE performance drops to $2.68 \pm 0.38$ from $0.29 \pm 0.11$ (MONODE + dynamic modulator). For rotating MNIST, MONODE + dynamics modulator performance drops to $0.096 \pm 0.008$ from $0.030 \pm 0.001$ (MONODE + static modulator). In addition, we examined the latent embeddings of the dynamics modulator for rotating MNIST. Where previously for the content modulator we observed clusters corresponding to a digit's class, for dynamics modulator such a topology in the latent space is not present (see App. D Fig. 17). Taken together with Table 3, the results confirm that the modulator variables are correlated with the true underlying factors of variation and play their corresponding roles.

### 5.4 Real world application: modulator variables

Next, we evaluate our framework on a subset of CMU Mocap dataset, which consists of 56 walking sequences from 6 different subjects. We pre-process the data as described in [Wang et al., 2007], resulting in 50-dimensional data sequences. We consider two data splits in which *(i)* the training and test subjects are the same, and *(ii)* one subject is reserved for testing. We refer to the datasets as MOCAP and MOCAP-SHIFT (see App. B for details). In test time, the model receives the first 75 observations as input and predicts the full horizon (150 time points). We repeat each experiment five times and report the mean and standard deviation of the MSEs computed on the full horizon, As shown in Table 4 and Fig. 5, our framework improves upon NODE. See Table 12 for ablations with different latent dimensionalities and with only one modulator variable (static or dynamics) present.

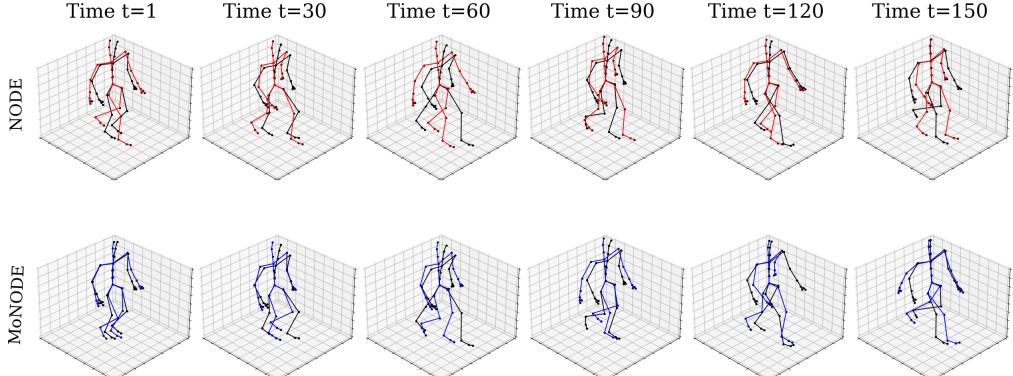

Figure 5: MoNODE and NODE reconstructions on a Mocap test sequence. Note that the models take the first $T_{\text{in}} = 75$ time points as input and predict for $T = 150$ time steps.

## 6    Discussion

The present work introduces a novel modulator framework for NODE models that allows to separate time-evolving ODE states from modulator variables. In particular, we introduce two types of modulating variables: *(i) dynamics modulator* that can modulate the dynamics function and *(ii) static modulator* that can modulate the observation mapping function. Our empirical results confirm that our framework improves generalization to new dynamics and far-horizon forecasting. Moreover, our modulator variables better capture the true unknown factors of variation as measured by $R^2$ score, and, as a result, the latent ODE states have an equivalent correspondence to the true observations.

**Limitations and future work**    The dynamical systems explored in the current work are limited to deterministic periodic systems that have different underlying factors of variation. The presented work introduces a framework that builds upon a base NODE model, hence, the performance of Mo*NODE is largely affected by the base model's performance. The current formulation cannot account for epistemic uncertainty and does not generalize to out-of-distribution modulators, because we maintain point estimates for the modulators. A straightforward extension would be to apply our framework to Gaussian process-based ODEs [Hegde et al., 2022] stochastic dynamical systems via an auxiliary variable that models the noise, similarly to Franceschi et al. [2020]. Likewise, rather than maintaining point estimates, the time-invariant modulator parameters could be inferred via marginal likelihood as in van der Wilk et al. [2018], Schwöbel et al. [2022], leading to a theoretically grounded Bayesian framework. Lastly, the separation of the dynamic factors and modulating factors could be explicitly enforced via an additional self-supervised contrasting loss term [Grill et al., 2020] or by more recent advances in representation learning [Bengio et al., 2013]. Lastly, the concept of *dynamics modulator* could also be extended to object-centric dynamical modeling [Kabra et al., 2021], which would allow accounting for per-object specific dynamics modulations while using a single dynamic model.

**Acknowledgements**    The data used in this project was obtained from `mocap.cs.cmu.edu`. The database was created with funding from NSF EIA-0196217. Çağatay Yıldız funded by the Deutsche Forschungsgemeinschaft (DFG, German Research Foundation) under Germany's Excellence Strategy – EXC-Number 2064/1 – Project number 390727645. This research utilized compute resources at the Tübingen Machine Learning Cloud, DFG FKZ INST 37/1057-1 FUGG. This project has received funding from the European Research Council (ERC) under the European Union's Horizon 2020 research and innovation programme (grant agreement No 950086).

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

# A   Model Encoder Adjustments

**SONODE**   In the original implementation by [Norcliffe et al., 2020] the initial velocity is computed by a 3-layer MLP with ELU activation functions and only the initial time point, $T_z = 1$, is passed as an input. For the initial position the first observation time point is passed, e.g. $\mathbf{x}_0$. The resulting model fails to fit on sinusoidal data, see fig. 6 for performance on validation trajectories.

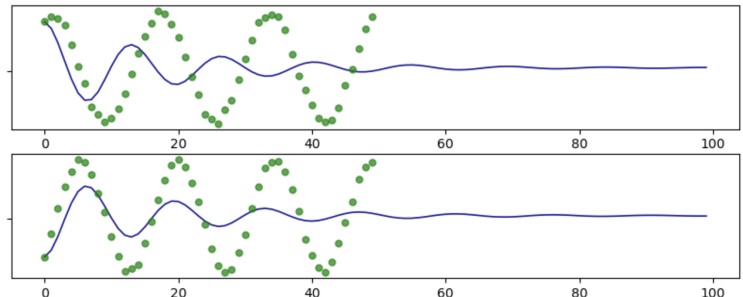

Figure 6: Original SONODE [Norcliffe et al., 2020] performance on sine validation sequences with $T_z = 1$. Green dots is ground truth, blue line, model's prediction.

As it can be seen in figure 6 the model only fits the first data point correctly, therefore, we extend the number of input time points passed to the model to $T_z = 5$ to compute the initial velocity. This results in notable performance improvements, see Fig. 7.

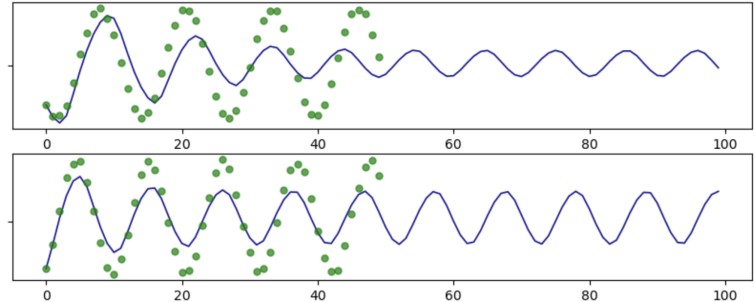

Figure 7: SONODE [Norcliffe et al., 2020] performance on sine validation sequences with $T_z = 5$. Green dots is ground truth, blue line, model's prediction.

Consequently, we increase the number of input frames used for SONODE, as well as investigate replacing the MLP architecture with a RNN. We report the resulting test MSEs in App. D Table 10. For the parameters used in the experiments please see Table 7.

**HBNODE**   In the original implementation by [Xia et al., 2021] the encoder is a 3-layer MLP with TANH activation functions that autoregressively takes every ground truth data point as input, e.g. $\mathbf{x}_i$ for $i = [0, \dots, T]$ and predicts the latent representation $\mathbf{z}_i$ which is also fed as input to the proceeding time points $i > 1$. The differential function is modeled by linear layer projection. As the present work is focused on model forecasting capabilities we adjust the original HBNODE to be compatible with problem set-ups where the ground truth data is not available. Meaning that once the model has reached the time point where there is no more ground truth data available ($T > 50$), the model only uses it's own latent state predictions $\mathbf{z}_i$ to compute the next latent state $\mathbf{z}_{i+1}$. As the authors of HBNODE claim that the model is designed to better fit long-term dependencies we initially reduce the number of subsequent increments, $N_{\text{incr}}$ from 10 to 3 (see App. B for training details). In Fig. 8 we show the qualitative performance of HBNODE on validation trajectories. Even though the model fits perfectly the ground truth data within $T < 50$, it fails to extrapolate to future time frames. We investigate the cause of this issue by, first, changing the differential function from a linear projection to a 3-layer MLP with TANH activations, and, second, by replacing the original autoregressive encoder with a RNN encoder that learns the initial position $\mathbf{z}_0$.

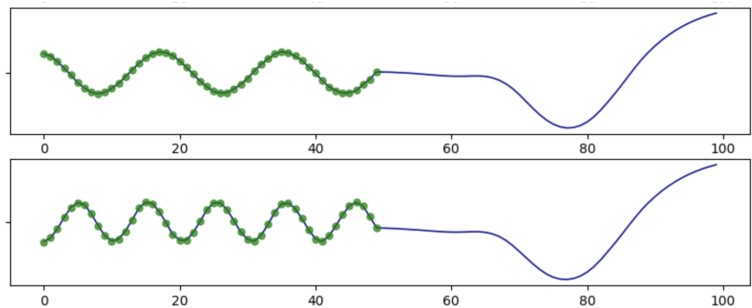

Figure 8: HBNODE [Xia et al., 2021] performance on sine validation sequences. Green dots is ground truth, blue line, model's prediction.

Table 5: Ablation results for HBNODE with RNN encoder, $T_z = 10$, with different $N_{incr}$ on sine data.

| $N_{incr}$ | Validation MSE | | | |
|---|---|---|---|---|
| | 1 | 3 | 5 | 10 |
| HBNODE | 0.164 | 0.135 | 0.144 | 0.066 |

Increasing the modelling capacity of the differential function did not resolve the observed issue with forecasting. This implies that the observed over-fitting of the model might be caused due to its autoregressive encoder that takes every ground truth time point as input. By replacing the autoregressive encoder with an RNN encoder we obtain significant forecasting improvements, see Fig. 9. Therefore, in all our reported experiments we replace the autoregressive encoder with an RNN and performed additional tests on the number of data increments $N_{incr} = [1, 3, 5, 10]$ as well as the number of input frames $Tiv$ to compute the initial state $\mathbf{z}_0$. We find in our experimentation that $N_{incr} = 10$ brings the best model performance (see Table 5) and subsequently test different $T_z$ frames, 10 and 40, respectively. We report the obtained test MSE for each set-up across 3 model runs with different seeds in App. D Table 10 and for the final model parameter settings used in the experiments please see App. C.

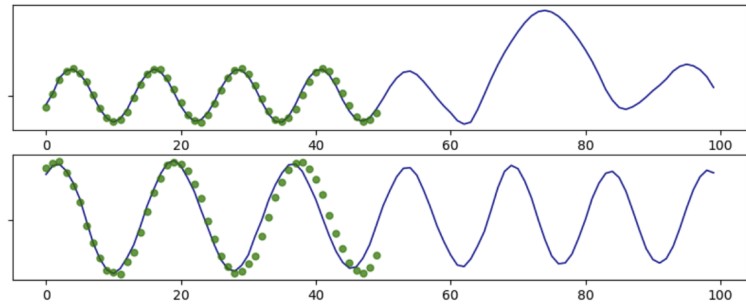

Figure 9: HBNODE [Xia et al., 2021] with RNN encoder, $N_{incr} = 10$ and $T_z = 10$ performance on sine validation sequences. Green dots is ground truth, blue line, model's prediction.

# B Experiment details

All models for sine, PP and rotating MNIST have been trained on a single GPU (GTX 1080 Ti) with 10 CPUs and 30G memory.

**Sine data** In the same vein as Rubanova et al. [2019], Norcliffe et al. [2021], we first define a set of time points $T = \{0, 0.1, 0.2, \ldots, t_i, \ldots, 4.9\}$. Then each sequence $\mathbf{y}^n$ is generated as follows:

$$a^n \sim \mathbb{U}[1, 3] \tag{15}$$
$$\omega^n \sim \mathbb{U}[0.5, 1.0] \tag{16}$$
$$\phi \sim \mathbb{U}[0, 1.0] \tag{17}$$
$$\epsilon^n \sim \mathbb{U}[0., \eta] \tag{18}$$
$$x_i^n = a^n \sin\left(\omega^n t_i + \phi^n\right) \tag{19}$$
$$y_i^n = x_i^n + \epsilon^n \tag{20}$$
$$\tag{21}$$

where $\mathbb{U}$ denotes the uniform distribution and $\eta$ is set to $\eta = 0.2$. We use $\Delta t = 0.1$ to generate $N = 300$ training trajectories of length $N_{\text{test}} = 50$, $N_{\text{val}} = 50$ validation trajectories of length $T_{\text{val}} = 50$, and $N_{\text{test}} = 50$ test trajectories of length $T_{\text{test}} = 150$. Test trajectories are longer to test model forecasting accuracy.

During training we set batch size to 10. We incrementally increase the sequence length of training trajectories starting from $T = 5$ until $T = 50$. We perform this in 10 increments for NODE, MoNODE, HBNODE and MoHBNODE, and in 4 increments for SONODE and MoSONODE. All models are trained for 600 epochs. The model with lowest validation MSE is used for evaluation on test trajectories.

**Predator-Prey data** To generate Predator-Prey trajectories we define the following generative process:

$$\alpha^n \sim \mathbb{U}[1., 3.5] \tag{22}$$
$$\gamma^n \sim \mathbb{U}[1., .3.5] \tag{23}$$
$$\delta^n \sim \mathbb{U}[.2, .3] \tag{24}$$
$$\beta^n \sim \mathbb{U}[.5, .6] \tag{25}$$
$$\epsilon^n \sim \mathbb{U}[0., \eta] \tag{26}$$
$$\mathbf{x}_0^n \sim \mathbb{U}[1., 5.] \tag{27}$$
$$\mathbf{x}_i^n = \mathbf{x}_0^n + \int_0^{t_i} \begin{bmatrix} \alpha^n x_1^n(\tau) - \beta^n x_1^n(\tau) x_2^n(\tau) \\ \delta^n x_1^n(\tau) x_2^n(\tau) - \gamma^n x_2^n(\tau) \end{bmatrix} \mathrm{d}\tau \tag{28}$$
$$\mathbf{y}_i^n = \mathbf{x}_i^n + \epsilon^n \tag{29}$$
$$\tag{30}$$

where $\mathbb{U}$ denotes the uniform distribution and $\eta$ is set to $\eta = 0.3$. We use $\Delta t = 0.1$ to generate $N = 600$ training trajectories of length $T = 100$, $N_{\text{val}} = 100$ validation trajectories of length $T_{\text{val}} = 100$, and $N_{\text{test}} = 100$ test trajectories of length $T_{\text{test}} = 300$. Test trajectories are longer to test model forecasting accuracy.

During training we set the batch size to 20. We incrementally increase the sequence length of training trajectories starting from $T = 10$ until $T = 100$. We perform this in 10 increments for NODE, MoNODE, HBNODE and MoHBNODE, and in 2 increments for SONODE and MoSONODE. All models are trained for 1500 epochs. The model with lowest validation MSE is used for evaluation on test trajectories.

**Rotating MNIST** For the data generation we base our code upon the image rotation implementation provided by [Solin et al., 2021]. We set the total number of rotation angles to $T = 16$ and sample the initial rotation angle from all the possible angles $\theta^n \sim \mathbf{U}[1, T]$. We generate $N = 1000$ training trajectories with length $T = 16$ and $N = 100$ validation and test trajectories with length $T = 45$.

During training we set the batch size to 25 and learning rate to 0.002. We train all models for 400 epochs and for evaluation use the trained model with the lowest validation mse.

**Bouncing ball with friction** For data generation, we use the script provided by Sutskever et al. [2008]. As a tiny update, each observed sequence has a friction constant drawn from a uniform distribution $\mathbb{U}[0, 0.1]$. We set the initial velocities to be unit vectors with random directions.

Table 6: Details of the datasets. Below, $N$ denotes the number of sequences (for training, validation, and test), $N_t$ is the sequence length, $\Delta t$ denotes the time between two observations, and $\sigma$ is the standard deviation of the added noise.

| DATASET | $N$ | $N_{\text{val}}$ | $N_{\text{test}}$ | $T$ | $T_{\text{val}}$ | $T_{\text{test}}$ | $\Delta t$ | $\eta$ |
|---|---|---|---|---|---|---|---|---|
| SINUSOIDAL DATA | 300 | 50 | 50 | 50 | 50 | 150 | 0.1 | 0.2 |
| LOTKA-VOLTERRA | 600 | 100 | 100 | 100 | 100 | 300 | 0.1 | 0.3 |
| ROTATING MNINST | 2000 | 100 | 100 | 16 | 45 | 45 | 0.1 | 0.0 |
| BOUNCING BALL | 1000 | 100 | 100 | 20 | 20 | 40 | 0.1 | 0 |

**Motion capture** The walking sequences we consider are `07_01.amc`, `08_03.amc`, `35_02.amc`, `35_12.amc`, `35_34.amc`, `39_08.amc`, `07_02.amc`, `16_15.amc`, `35_03.amc`, `35_13.amc`, `38_01.amc`, `39_09.amc`, `07_03.amc`, `16_16.amc`, `35_04.amc`, `35_14.amc`, `38_02.amc`, `39_10.amc`, `07_06.amc`, `16_21.amc`, `35_05.amc`, `35_15.amc`, `39_01.amc`, `39_12.amc`, `07_07.amc`, `16_22.amc`, `35_06.amc`, `35_28.amc`, `39_02.amc`, `39_13.amc`, `07_08.amc`, `16_31.amc`, `35_07.amc`, `35_29.amc`, `39_03.amc`, `39_14.amc`, `07_09.amc`, `16_32.amc`, `35_08.amc`, `35_30.amc`, `39_04.amc`, `07_10.amc`, `16_47.amc`, `35_09.amc`, `35_31.amc`, `39_05.amc`, `07_11.amc`, `16_58.amc`, `35_10.amc`, `35_32.amc`, `39_06.amc`, `08_02.amc`, `35_01.amc`, `35_11.amc`, `35_33.amc`, `39_07.amc`. The number of training, validation and test samples in MOCAP and MOCAP-SHIFT splits are 46-5-5 and 43-5-8, respectively. Since sequences are already dense, we skip every other data point. For ease of implementation, we take the last 300 time points, leading to sequences of length $T = 150$. We experiment with different latent dimensionalities and report the findings in Table 12.

# C   Architecture and Hyperparameter Details

Table 7: Experiment parameters. $T_z$ is the number of input frames to condition the initial state on, and $T_e$ is the number of input frames used to compute the modulator/content variables. Further, we note the dimensionality of the dynamics state $q_z$ (output of the ODESolve, where the differential equation is modeled by an MLP), as well as the latent dimensionality of the dynamics modulator $q_d$ and the static $q_s$. lr stands for learning rate.

| DATASET | MODEL | $T_z$ | $T_e$ | $q_z$ | $q_d$ | $q_s$ | lr | N# parameters |
|---|---|---|---|---|---|---|---|---|
| | NODE | 10 | - | 8 | - | - | 0.002 | 24666 |
| | MONODE | 3 | 10 | 4 | 4 | - | 0.002 | 24598 |
| SIN | SONODE | 10 | - | 2 | - | - | 0.002 | 21802 |
| | MOSONODE | 10 | 10 | 2 | 4 | - | 0.002 | 23346 |
| | HBNODE | 10 | - | 8 | - | - | 0.002 | 24404 |
| | MOHBNODE | 10 | 10 | 4 | 4 | - | 0.002 | 24938 |
| | NODE | 40 | - | 16 | - | - | 0.002 | 28022 |
| | MONODE | 8 | 40 | 8 | 8 | - | 0.002 | 26976 |
| PREY-PREDATOR | SONODE | 40 | - | 4 | - | - | 0.002 | 29204 |
| | MOSONODE | 40 | 40 | 4 | 8 | - | 0.002 | 31382 |
| | HBNODE | 40 | - | 16 | - | - | 0.002 | 26586 |
| | MOHBNODE | 10 | 40 | 8 | 8 | - | 0.002 | 26744 |
| ROTATING MNIST | NODE | 5 | - | 32 | - | - | 0.001 | 513561 |
| | MONODE | 5 | 15 | 16 | 16 | - | 0.001 | 558681 |
| MOCAP | NODE | 75 | - | 24 | - | - | 0.002 | 45112 |
| | MONODE | 75 | 75 | 8 | 8 | 8 | 0.002 | 51360 |
| MOCAP-SHIFT | NODE | 75 | - | 24 | - | - | 0.002 | 45112 |
| | MONODE | 75 | 75 | 8 | 8 | 8 | 0.002 | 51360 |
| BOUNCING BALL | NODE | 5 | - | 212 | - | - | 0.001 | 162343 |
| | MONODE | 5 | 5 | 8 | 4 | - | 0.001 | 150479 |

Table 8: Model architecture overview

| Dataset | Model | Position Encoder | Velocity Encoder | Modulator P. Network | Differential Function | ODE Solver | Decoder |
|---|---|---|---|---|---|---|---|
| SIN | NODE | RNN | - | - | MLP | rk4 | MLP |
| PREY-PREDATOR | MONODE | RNN | - | RNN | MLP | rk4 | MLP |
| MOCAP | SONODE | - | MLP | - | MLP | rk4 | - |
| MOCAP-SHIFT | MOSONODE | - | MLP | RNN | MLP | rk4 | - |
| ROTATING MNIST | NODE | CNN | - | - | MLP | rk4 | CNN |
| | MONODE | CNN | - | RNN | MLP | rk4 | CNN |
| BOUNCING BALL | NODE | CNN | CNN | - | MLP | rk4 | CNN |
| | MONODE | CNN | CNN | CNN | MLP | rk4 | CNN |

# D  Additional Results

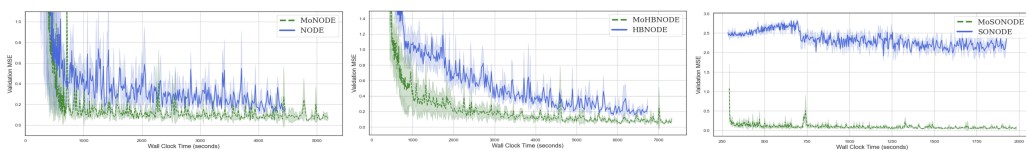

Figure 10: Model training time comparison on sinusoidal data. x-axis validation MSE, y-axis wall clock time in seconds. Thick line mean value across 3 different runs (different initalization seeds), shaded area standard deviation.

Table 9: Test MSE across different solvers for sinusoidal data for NODE Chen et al. [2018] and MONODE (ours).

| model | seq. length | euler | rk4 | dopri5 |
|---|---|---|---|---|
| NODE | $T = 50$ | 0.13( 0.03) | 0.13 (0.03) | 0.19( 0.09) |
| MONODE | | **0.07( 0.02)** | **0.04 (0.01)** | **0.05( 0.01)** |
| NODE | $T = 150$ | 2.48( 1.26) | 1.84 (0.70) | 2.11( 0.56) |
| MONODE | | **0.42( 0.21)** | **0.29 (0.11)** | **0.31( 0.04)** |

Table 10: Ablation results for different model architectures and hyperparameters for Sinusoidal and Predator-Prey data. For final model parameters used see section app. C

| Data | Model | Velocity Encoder | $T_z$ | $T_e$ | $N_{incr}$ | Test MSE (std) $T = 50$ | $T = 150$ |
|---|---|---|---|---|---|---|---|
| SINUSOIDAL DATA | MONODE | - | 3 | 10 | 10 | 0.04 ( 0.01) | 0.29 ( 0.11) |
| | | - | 10 | 10 | 10 | 0.05 ( 0.01) | 0.37 ( 0.12) |
| | SONODE | MLP | 5 | - | 10 | 2.18 ( 0.06) | 3.05 ( 0.02) |
| | | MLP | 10 | - | 4 | 1.81 ( 0.01) | 3.05 ( 0.07) |
| | | RNN | 10 | - | 10 | 2.19 ( 0.15) | 3.05 ( 0.07) |
| | MOSONODE | MLP | 5 | 10 | 10 | 0.04 ( 0.00) | 0.29 ( 0.04) |
| | | MLP | 10 | 10 | 4 | 0.04 ( 0.00) | 0.32 ( 0.06) |
| | | RNN | 10 | 10 | 10 | 0.05 ( 0.01) | 0.35 ( 0.10) |
| | | | | | | $T = 100$ | $T = 300$ |
| PREDATOR-PREY | MONODE | - | 8 | 40 | 10 | 0.74 ( 0.05) | 4.33 ( 0.19) |
| | | - | 40 | 40 | 10 | 0.57 ( 0.04) | 23.05 ( 1.49) |
| | SONODE | MLP | 10 | - | 10 | 15.801 (0.748) | 40.921 (0.816) |
| | | MLP | 40 | - | 2 | 5.130 (0.221) | 39.260 (3.367) |
| | | RNN | 40 | - | 10 | 5.101(0.339) | 41.890 (8.444) |
| | MOSONODE | MLP | 10 | 40 | 10 | 1.459 (0.284) | 6.695 (0.828) |
| | | MLP | 40 | 40 | 2 | 1.093 (0.059) | 6.342 (0.982) |
| | | RNN | 40 | 40 | 10 | 1.294 (0.322) | 8.639 (1.468) |
| | HBNODE | RNN | 40 | - | 10 | 0.879 (0.096) | 3346.625 (2119.239) |
| | | RNN | 10 | - | 10 | 13.803 (0.138) | 2833.051 (3254.703) |
| | MOHBNODE | RNN | 40 | 40 | 10 | 0.870 (0.088) | 225.012 (274.485) |
| | | RNN | 10 | 40 | 10 | 0.943 (0.092) | 10.205 (1.429) |

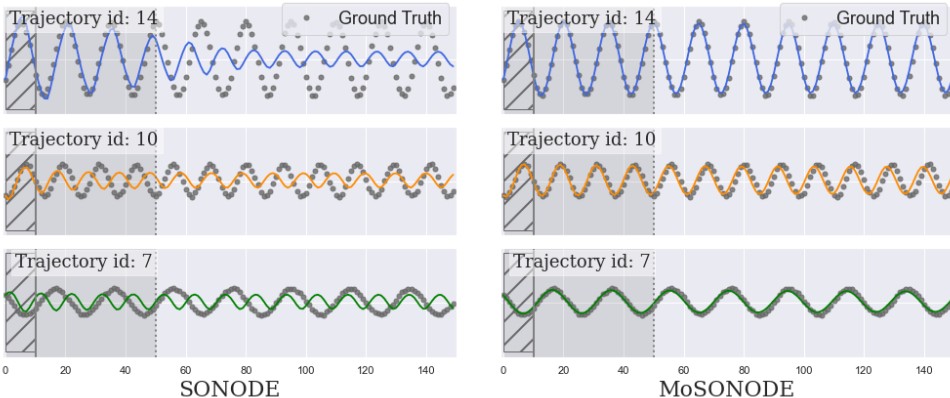

Figure 11: Qualitative performance improvement by MOSONODE on SONODE on sinusoidal data. Both models have the same architecture: MLP Velocity encoder, $T_z = 5$, and $N_{incr} = 4$. Lined area represents number of input frames used to compute initial position and velocity, while grey area represent number of frames seen during training.

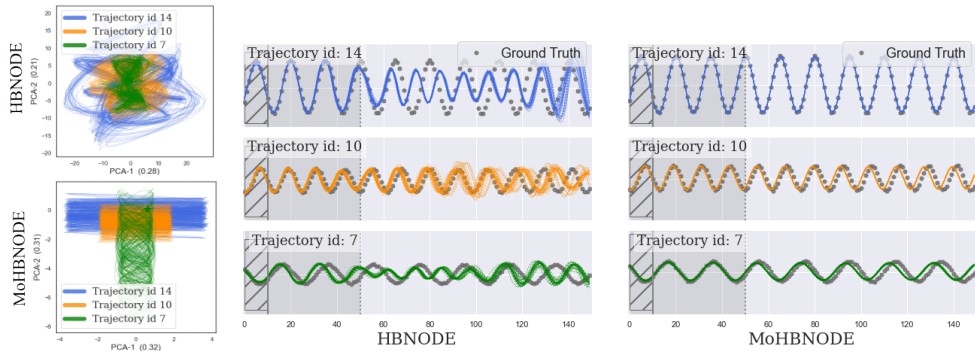

Figure 12: The left panel illustrates PCA embeddings of the dynamics state inferred by HBNODE and MOHBNODE, where each color matches one trajectory. The figures on right compare HBNODE and MOHBNODE reconstructions on test trajectory sequences. Note that the models take the dashed area as input, and darker area represents the trajectory length seen during training.

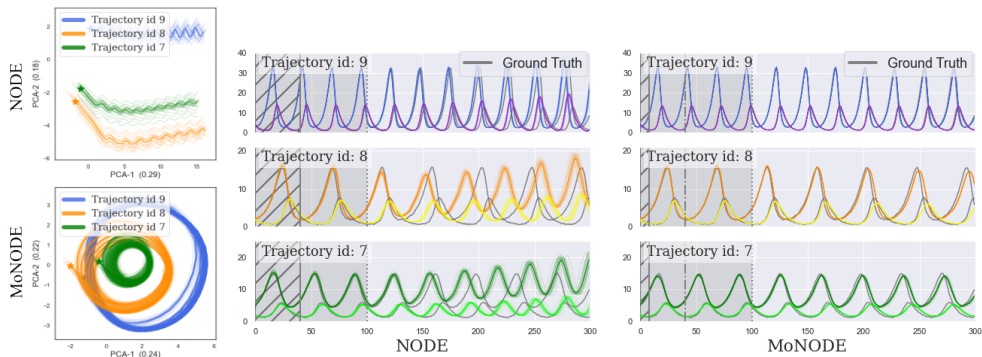

Figure 13: The left panel illustrates PCA embeddings of the dynamics state inferred by NODE and MONODE, where each color matches one trajectory. The figures on right compare NODE and MONODE reconstructions on test trajectory sequences. Note that the models take the dashed area as input, and darker area represents the trajectory length seen during training. Prey-Predator data.

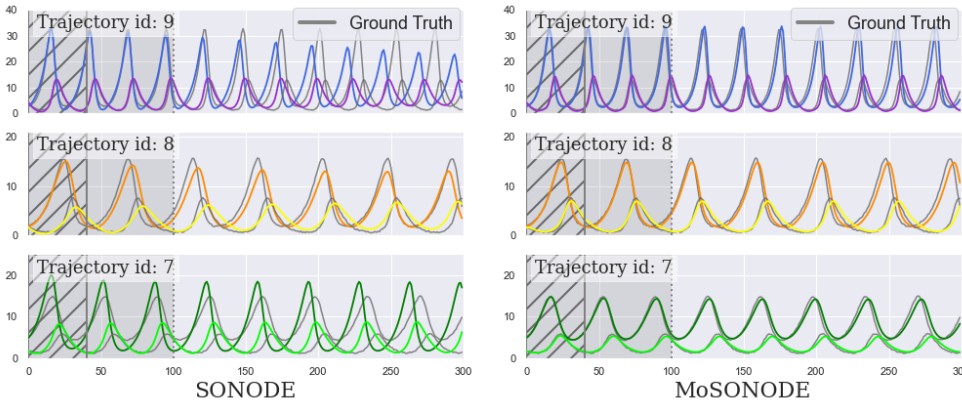

Figure 14: Qualitative performance improvement by MOSONODE on SONODE on Predator-Prey data. Both models have the same architecture: MLP Velocity encoder, $T_z = 40$, and $N_{\text{incr}} = 2$. Lined area represents number of input frames used to compute initial position and velocity, while grey area represent number of frames seen during training.

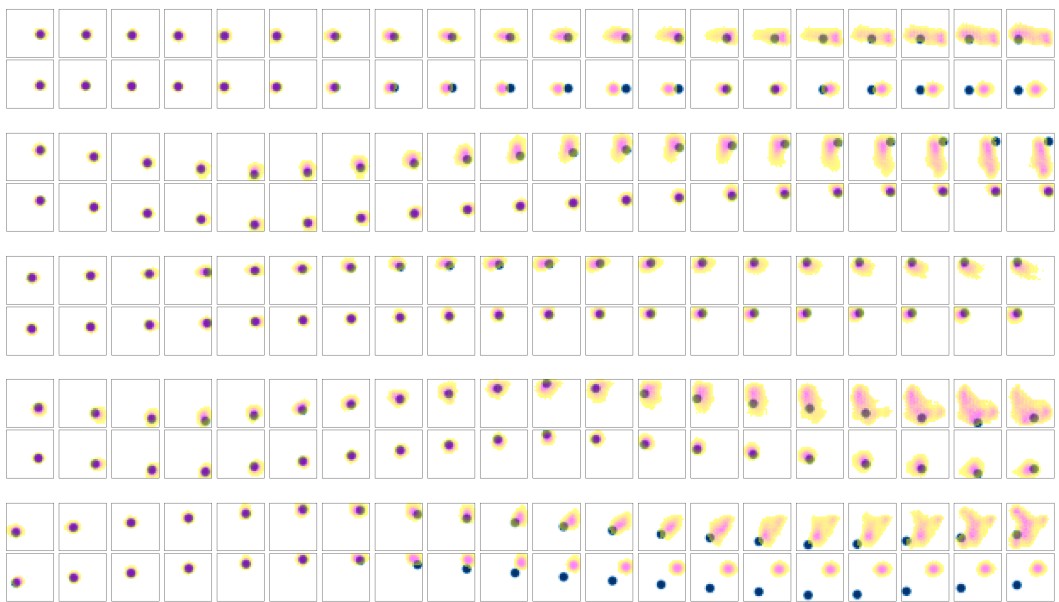

Figure 15: NODE and MONODE predictions on five bouncing ball sequences. For each couple of plots, the first and second rows correspond to NODE and MONODE predictions, respectively. The dark circles are the ground truth ball positions and the colored background is the model prediction. As can be seen, MONODE tracks the ball much better compared to NODE baseline.

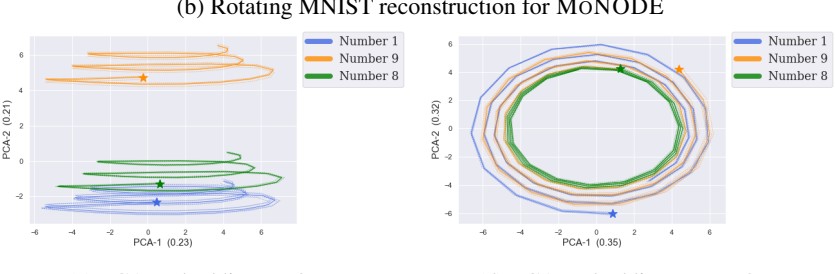

(a) Rotating MNIST reconstruction for NODE

(b) Rotating MNIST reconstruction for MoNODE

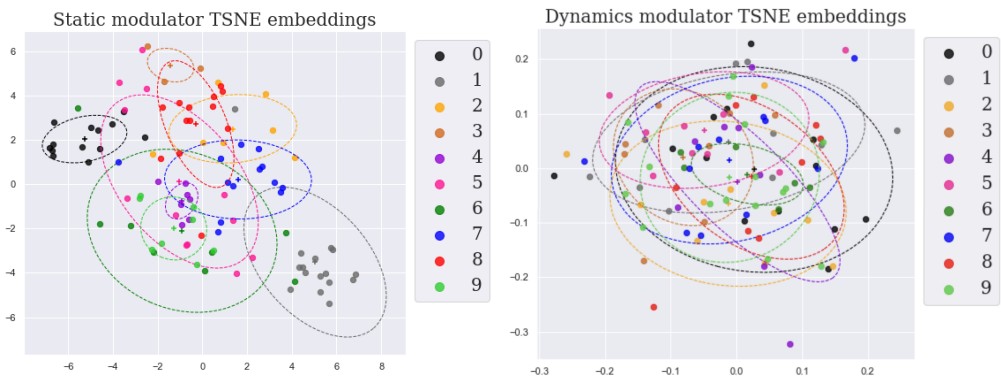

(c) PCA embeddings NODE        (d) PCA embeddings MoNODE

Figure 16: Top two figures are (a) NODE and (b) MoNODE reconstruction of 3 different test trajectories, where top line is the ground truth for every digit. Bottom row illustrates the corresponding PCA embeddings of each trajectory plotted above, where (c) corresponds to NODE and (d) corresponds to MoNODE.

Figure 17: (left) TSNE embeddings of the static modulator as inferred by MoNODE on test data for roating MNIST. Each color represent one of the possible 10 digits. For digits that have a distinct shape, such as "0" and "1", we can discern a clear clustering and separation from the other digits, while for the remaining digits the per digit clusters exhibit more overlap. This is to be expected as we do not constrain the model to learn spatially separated clusters, as well as some digits are more similar either due to their appearance when rotated, such as "6" and rotated "9", or due to their writing style. (right) For further illustration, we repeat the same experiment with dynamics modulators instead of static. Unlike static modulator latents, for dynamics modulator no topology in the latent space is not present. In all, the results confirm that the modulator variables are correlated with the true underlying factors of variation

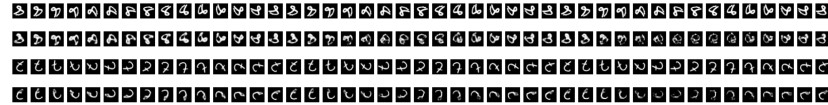

(a) Rotating MNIST reconstruction for NODE

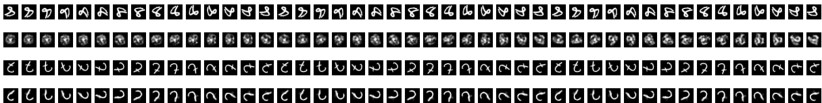

(b) Rotating MNIST reconstruction for MONODE

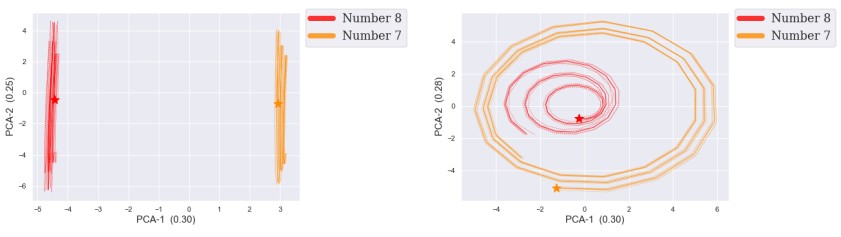

(c) PCA embeddings NODE  (d) PCA embeddings MONODE

Figure 18: Failure cases on Rotating MNIST for (a) NODE and (b) MONODE reconstruction of 2 different test trajectories, where top line is the ground truth for every digit. Bottom row illustrates the corresponding PCA embeddings of each trajectory plotted above, where (c) corresponds to NODE and (d) corresponds to MONODE. Despite MONODE failing to reconstruct the correct shape of the digit the latent PCA embedding plots reveal that the model has still captured the circular rotation motion.

Table 11: Ablation studies on Rotating MNIST data to compare different parameter affects on NODE and MONODE. As it can be noted the test MSE for reconstruction within training sequence length T=15 is lower for NODE than for MONODE, while for forecasting MONODE outperforms NODE. This discrepancy can be explained by the fact that the current results indicate that MONODE is harder to optimize as it's test mse can be grouped in two modes: success cases were the reconstructions are great and the error remains low for forecasting 16 or failure cases where the model fails to reconstruct already within the training sequence length T=15 as in fig 18. This explains the higher test MSE for MONODE at $T = 15$ and it remaining almost unchanged for $T = 45$, while for NODE the test MSE grows as the sequence length increases.

| model | $q_z$ | $q_s$ | #N parameters | Test MSE (std) $T = 15$ | Test MSE (std) $T = 45$ |
|---|---|---|---|---|---|
| NODE | 16 | - | 461161 | 0.020 | 0.098 |
| MONODE | 6 | 10 | 513637 | 0.031 | 0.034 |
| | 8 | 8 | 516089 | 0.031 | 0.032 |
| NODE | 24 | - | 487361 | 0.014 | 0.062 |
| MONODE | 8 | 16 | 532481 | 0.030 | 0.031 |
| | 12 | 12 | 537385 | 0.035 | 0.038 |
| NODE | 32 | - | 513561 | 0.013 | 0.042 |
| MONODE | 8 | 24 | 548873 | 0.035 | 0.037 |
| | 12 | 20 | 553777 | 0.033 | 0.033 |
| | 16 | 16 | 558681 | 0.031 | 0.032 |

Table 12: Ablation studies on mocap data to compare NODE and different variants of our model (with/without dynamics modulator and content variable). We also experiment with different latent dimensionalities.

| | $q_z$ | $q_d$ | $q_s$ | MOCAP | MOCAP-SHIFT |
|---|---|---|---|---|---|
| NODE | 6 | - | - | $61.6 \pm 9.1$ | $81.6 \pm 39.5$ |
| | 12 | - | - | $71.2 \pm 20.0$ | $63.1 \pm 8.0$ |
| | 24 | - | - | $72.2 \pm 12.4$ | $61.6 \pm 6.2$ |
| MONODE | 3 | 3 | - | $76.4 \pm 12.7$ | $72.6 \pm 6.2$ |
| | 6 | 6 | - | $79.7 \pm 14.4$ | $83.2 \pm 23.0$ |
| | 12 | 12 | - | $66.7 \pm 12.3$ | $78.9 \pm 12.7$ |
| | 3 | - | 3 | $72.6 \pm 7.9$ | $67.0 \pm 18.4$ |
| | 6 | - | 6 | $64.6 \pm 9.3$ | $63.9 \pm 16.2$ |
| | 12 | - | 12 | $60.0 \pm 17.6$ | $62.5 \pm 11.0$ |
| | 2 | 2 | 2 | $79.2 \pm 8.7$ | $56.3 \pm 7.8$ |
| | 4 | 4 | 4 | $78.8 \pm 16.7$ | $57.0 \pm 7.8$ |
| | 8 | 8 | 8 | $57.7 \pm 9.8$ | $58.0 \pm 10.7$ |

