# OpenReview forum: "Modulated Neural ODEs"
_NeurIPS.cc/2023/Conference — NeurIPS 2023 poster_

### Official Review · Reviewer_Znd9 · 2023-07-06

**Soundness:** 3 good
**Presentation:** 3 good
**Contribution:** 3 good
**Rating:** 6
**Confidence:** 3

**Summary:**

This paper proposes a modulated neural ODE (MoNODE) that can separately model the dynamics and time-invariant modulating variables (e.g., parameters of dynamics or contents of agents).  Such modulating variables are modeled by averaging the embedded trajectories of given observations (i.e., the outputs of modulator prediction networks for a observed trajectories) with respect to the time, similar to the tranformation-invariant (TI) pooling approach. The proposed approach can be integrated into various NODE models simply. The authors validate the proposed MoNODE for various dynamical systems, including some synthetic examples (sinusoidal, Lotka–Volterra Hamiltonian, bouncing ball), images dynamics (rotated MNIST), and CMU motion capture dataset. The experimental results demonstrate the proposed MoNODE is better than NODE in terms of the interpretablility well as long-term predictability of dynamics.

***

This paper proposes a concept of separating and learning dynamics and time-invariant modulator variables, and I believe it could be beneficial for various time-series prediction problems in scientific and engineering fields. However, I have doubts about whether the proposed model maintains the time-invariance property explicitly. Additionally, there are some concerns regarding comparisons with similar papers and validation in more complicated chaotic systems. Therefore, I currently consider this paper to be at a borderline reject.

**Strengths:**

- The proposed approach is simple and principled, making it easy to integrate into various NODE-type models.

- The paper is well-written and easy to follow.

- The experimental results demonstrate that the learned latent dynamics of MoNODEs are more convincing and interpretable compared to those of NODEs. This may be attributed to the explicit decomposition of the time-invariant contents of agents from the dynamical behavior itself.

**Weaknesses:**

- The time-invariance of the moduator prediction network is only gauranteed for the finite and fixed time horizon of $T_e$. As a result, it is not entirely clear whether the proposed model can effectively capture the time-invariant aspects of observations over an arbitrary time span.

- Similarly, the proposed model requires the observation of length $T_e$ to predict the time-invariant modulating variables. Therefore, it is not greatly applicable for the case when $T_z < T_e$.

- The paper lacks a comparison with a relevant previous work [1], which factorizes the latent space into content and motion variables and models the dynamics using the (Hamiltonian) latent NODE.


***

[1] Khan, A., & Storkey, A. J. Hamiltonian latent operators for content and motion disentanglement in image sequences. NeurIPS 2022.

**Questions:**

- I think it would be great to experimentally demonstrate that a trained modulating prediction network indeed guarantees time-invariance in the long-term time horizon. For example, it would be helpful to plot a graph to visualize the change in $d$ or $s$ over time (ideally to be a constant line).

- Is the performance of the trained MoNODE still good when $T_z = T_e \ll T_{train}$? It seems that an ablation study is needed to check this.

- I am curious whether MoNODE would be applicable to chaotic dynamics. For instance, one could consider the parameters ($\sigma$, $\rho$, $\beta$) of the Lorenz system as dynamic modulating variables. Can MoNODE effectively learn and infer them?

- Max pooling [2] is commonly used as the pooling function for invariant mappings (e.g., $s^n = \max s_{i}^{n}$ for the static modulator case). Is there a specific reason for using averaging? It would be nice if the authors could conduct a comparison between max and average pooling invariant maps.

***

[2] Laptev, D., Savinov, N., Buhmann, J. M., & Pollefeys, M. Ti-pooling: transformation-invariant pooling for feature learning in convolutional neural networks. CVPR 2016.

**Limitations:**

The authors mention some limitations (limited to deterministic systems, generalization to out-of-distribution modulators, ...) of the proposed method in the Discussion section. Another limitation worth mentioning is the weakness section of my review: not entirely guaranteed time-invariance, limited application when $T_z < T_e$, and the missing relevant reference.

---

> ### Author Rebuttal · Authors · 2023-08-09
>
> Thank you for the detailed review, please find below our response to the concerns voiced.
> > I think it would be great to experimentally demonstrate that a trained modulating prediction network indeed guarantees time-invariance in the long-term time horizon. For example, it would be helpful to plot a graph to visualize the change in or over time (ideally to be a constant line).
>
> We would like to clarify that the modulator prediction network does not guarantee time-invariance with respect to the predicted dynamic state $z$, but rather the modulator variables $d$ and $s$. Our proposed framework accumulates prediction error as commonly known with auto-regressive frameworks, yet minor compared to the existing base models (NODE, SONODE, HBNODE, LSONODE) as shown in Figures 2-3 and Figures 10-15 (Supp. mat. D) and indicated by the accuracy improvement with respect to test MSE: Sin by 84.47 \%, PP by 88.38 \%, BB by 20.12 \%, Rot.MNIST by 23\%, and MOCAP by 20.08\%.
>
> > Is the performance of the trained MoNODE still good when $T_z = T_e << T_{train}$ It seems that an ablation study is needed to check this.
>
> In general, the choice of $T_e$ (number of timepoints used for learning modulator variables) is independent of $T_{train}$ (total training sequence length) but rather depends on the underlying dynamical system's properties. Specifically, a subsequence of length $T_e$ has to have sufficient information of the change in dynamics to correctly estimate the system parameters. Hence, if the underlying dynamics are changing very slowly, a higher number of timepoints (larger $T_e$) are needed to estimate the underlying parameters.
> Secondly, the performance of MoNODE is good when $T_z = T_e$. As reported in Table 7 (Supp. mat., C) for most datasets these two parameters are matched and additional ablations are reported in Table 10 (Supp. mat. D).
> Lastly, we have paid detailed attention to match $T_z$ of NODE with $T_e$ of Mo-*NODE to make sure that the performance boost is not simply caused by a longer input sequence.
>
> > similarly, the proposed model requires the observation of length $T_e$ to predict the time-invariant modulating variables. Therefore, it is not greatly applicable for the case when $T_z < T_e$.
>
> Our framework is applicable to settings when $T_z < T_e$. Please see Table 10. (Supp. mat. D), where our framework outperforms base models also when $T_z < T_e$.
>
> > I am curious whether MoNODE would be applicable to chaotic dynamics. For instance, one could consider the parameters of the Lorenz system as dynamic modulating variables. Can MoNODE effectively learn and infer them?
>
> We perform an additional ablation on a Lorenz Attractor dataset, with varying parameters as suggested. MoNODE outperforms NODE on both short- and long-horizon forecasting. Specifically, the test MSEs for NODE and MONODE are 36.19 vs 33.90 for short term and 61.28 vs 46.56 for long term. We would like to remind that NODEs are known to suffer from solver stability issues on chaotic systems [4].
>
> [4] Allauzen, A., Dardis, T. P. M., & Plath, H. (2022). Experimental study of Neural ODE training with adaptive solver for dynamical systems modeling. arXiv preprint arXiv:2211.06972.
>
> > Max pooling [2] is commonly used as the pooling function for invariant mappings (e.g., for the static modulator case). Is there a specific reason for using averaging? It would be nice if the authors could conduct a comparison between max and average pooling invariant maps.
>
> We performed an additional ablation with Max pooling [2] operator instead of averaging for the static modulator on the rotating MNIST experiment and for the dynamics modulator on the sinusoidal data experiment. We report the obtained test MSE across 3 runs with different initialization seeds and corresponding standard deviation. For MoNODE + max pooling on rotating MNIST we obtain test MSE of 0.035 (0.004), which is  better than the baseline model NODE 0.039 (0.003) but is worse than MoNODE + averaging 0.030 (0.001). For MoNODE + max pooling on sinusoidal data we obtain test MSE of 0.28 (0.03), which is on par with MoNODE + averaging 0.29 (0.11) and better than the base model 1.84 (0.70). We chose the averaging operator in our work as modulator variables should capture features that remain constant in time, however, other functions are also a viable option, for example, one could also consider summation [3].
>
> [3] Franceschi, J. Y., Delasalles, E., Chen, M., Lamprier, S., & Gallinari, P. (2020, November). Stochastic latent residual video prediction. In International Conference on Machine Learning (pp. 3233-3246). PMLR.
>
> >  The paper lacks a comparison with a relevant previous work [1], which factorizes the latent space into content and motion variables and models the dynamics using the (Hamiltonian) latent NODE. ([1] Khan, A., \& Storkey, A. J. Hamiltonian latent operators for content and motion disentanglement in image sequences. NeurIPS 2022.)
>
> The code base for [1] unfortunately is incomplete. However, in Section 3.5 of [1] they also perform an experiment on Rotating MNIST (single digit), where ODE2VAE outperforms their method. In our work, we also compare our framework to the non-Bayesian version of ODE2VAE under the name of LSONODE (Section 5.1). Therefore, we perform an additional ablation of ODE2VAE performance on our implementation of Rotating MNIST experiment (all digits) with MoODE2VAE. We report the mean across 3 runs with std. The test MSE for ODE2VAE is 0.068 (0.002), while MoODE2VAE obtains 0.035 (0.002) outperforming ODE2VAE by a large margin. As such, we are confident to say that our method improves the current state-of-the-art and we will contact authors of [1] to include a complete comparison in the camera ready version.

---

> > ### Comment · Reviewer_Znd9 · 2023-08-21
> >
> > I appreciate the authors’ thorough response and clarification to my concerns. Specifically,
> >
> > **Time invariance of $d$ and $s$.**
> >
> > I understand the modulator variables $d$ and $s$ should be time-invariance, thus want to see whether $d$ and $s$ are constant over time, as I stated in my initial review. Since the review process does not allow for additional image uploads, it would be beneficial if the authors could include the corresponding results in the revised version of their manuscript.
> >
> > **$T_z$ and $T_e$.**
> >
> > Thank you for providing clarification on this matter. I have checked the Appendix as suggested by the authors, and am generally convinced.
> >
> > **Chaotic systems.**
> >
> > Thank you for the additional result. Although the result is not very strong, MoNODE outperforms NODE, thus might be potentially useful for this perspective.
> >
> > **Comparison with the relevant work.**
> >
> > Thank you for presenting the new observation. Although it is not a direct comparison with the reference I mentioned, the logic and outcomes that the authors provided are convincing to me.
> >
> > Overall, I think the merits of this paper outweigh my initial concerns, and the paper is worthy of publication. I raised my score accordingly.

---

### Official Review · Reviewer_Y7gY · 2023-07-09

**Soundness:** 3 good
**Presentation:** 2 fair
**Contribution:** 2 fair
**Rating:** 4
**Confidence:** 5

**Summary:**

This paper proposes a framework for Neural ODEs called Modulated Neural ODEs, which combines the conventional latent Neural ODEs with two Modulator Prediction Networks that are used to extract the time-invariant static and dynamic modulator variables, respectively. Their empirical results demonstrate that this framework can improve generalizations to new dynamic parameterization and long-time forecasting, and can better capture the true unknown factors of variation as well.

**Strengths:**

1.	The proposed MoNODE framework has better capability of generalizing to new dynamic parameterizations and long-term forecasting.

2.	They experimentally verify that the MoNODE framework is able to better capture the true unknown factors of variation with modulator variables.


**Weaknesses:**

1.	Experimental examples are not convincing enough. The dynamical systems chosen in this paper are relatively simple and naive. For more complex systems such as trajectories generated by a family of chaotic systems controlled by different parameters, it is doubtful whether the MoNODE framework can still extract effective modulator variables.

2.	Though NODE as the baseline group maintains the same dimension as MoNODE, additional parameters are still introduced in MoNODE, such as parameters in the Modulator Prediction Networks. It is not clear whether the slight improvement in performance is due to the additional number of parameters.

3.	This paper does not directly provide a visual presentation or intuitive explanation of the static and dynamics modulator variables extracted by the modulator prediction networks.


**Questions:**

1.	When comparing with NODE, could you provide a comparison of the number of parameters and set up a more reasonable experiment to prove that the performance improvement is not caused by the increase in the number of parameters?

2.	Whether the MoNODE framework can outperform the conventional NODE on the task of learning more complex dynamical systems, such as trajectories generated by a family of chaotic systems controlled by different parameters?

3.	For the CMU Mocap datasets, due to the addition of the modulator variables, the MoNODE is supposed to significantly improve the performance of NODE in the Mocap-Shift task, but in fact, the improvement effect is not obvious, which is far less than that of Mocap. Can you further explain this phenomenon?

4.	Could you directly provide some visual description and intuitive interpretation of the modulator variables extracted by the modulator prediction networks?


**Limitations:**

The authors have adequately addressed the limitations.

---

> ### Author Rebuttal · Authors · 2023-08-09
>
> Thank you for the detailed review. Please find below our response to the concerns voiced.
>
> > Experimental examples are not convincing enough.
>
> We are slightly surprised and sorry to hear that our findings fail to convince the reviewer. We exhaustively test our ideas on four NODE variants (standard, SONODE, HBNODE, LSONODE) on five datasets that were all previously used to evaluate similar approaches. Our Mo*NODE framework improved by 88.38\% test MSE for PP experiment and on average by 55.25\% across all experiments, as reported in Table 2. and Table 4. That being said, we would be happy to know what other experiments would help us convince the reviewer.
>
> > When comparing with NODE, could you provide a comparison of the number of parameters and set up a more reasonable experiment to prove that the performance improvement is not caused by the increase in the number of parameters?
>
> Thank you for voicing this concern, as indeed one candidate explanation for the observed performance improvement could be increased parameter count. To account for this fact we approximately match the number of parameters in the base network and the Modulated counterpart. For a complete overview of the number of parameters of each model for each experiment please see Table 7 in the supplementary material, Section C, Architecture and Hyperparameter Details. In particular, for sin, PP, and BB, NODE model actually has a higher parameter count than its MoNODE counterpart (We also control overfitting by choosing the best model via cross-validation). Furthermore, to also account for the fact that the performance boost is not caused just by a larger latent dimensionality, we match the latent dimensionality between the base model and the modulated counterpart. As such, we are confident to say that the observed performance boost is not caused simply by an increase in number of parameters, but rather by the framework itself.
>
> > Whether the MoNODE framework can outperform the conventional NODE on the task of learning more complex dynamical systems, such as trajectories generated by a family of chaotic systems controlled by different parameters?
>
> We perform an additional ablation on a Lorenz Attractor dataset, with varying parameters as suggested. MoNODE outperforms NODE on both short- and long-horizon forecasting. Specifically, the test MSEs for NODE and MONODE are 36.19 vs 33.90 for short term and 61.28 vs 46.56 for long term. We would like to remind that NODEs are known to suffer from solver stability issues on chaotic systems [1].
>
> [1] Allauzen, A., Dardis, T. P. M., & Plath, H. (2022). Experimental study of Neural ODE training with adaptive solver for dynamical systems modeling. arXiv preprint arXiv:2211.06972.
>
> > For the CMU Mocap datasets, due to the addition of the modulator variables, the MoNODE is supposed to significantly improve the performance of NODE in the Mocap-Shift task, but in fact, the improvement effect is not obvious, which is far less than that of Mocap. Can you further explain this phenomenon?
>
> This outcome can be attributed to the chosen data split. In particular, we chose to leave a particular subject out for testing since the other subjects had either too many or too few sequences (notice that leaving out a subject with too many/few sequences would lead to a huge decrease in the training data size or biased testing). Thanks for raising your concern, we will include a detailed explanation for this situation in our paper.
>
> > Could you directly provide some visual description and intuitive interpretation of the modulator variables extracted by the modulator prediction networks?
>
> Dynamics modulators: intuitively they capture information about the true underlying differential equation parameters (experiments sin, PP, BB) or in the case when the dynamic's function in closed form is unknown (Mocap), then the parameters that affect this unknown function (length of legs).
>
> To confirm that this intuitive explanation indeed holds, we computed the $R^2$ scores by performing regression from latent variables to the true parameters, as described in Schott et al. 2021. The regression input for MoNODE are the dynamics modulators, while for NODE it is the latent trajectories. For all experiments in Section 5.1, sinusoidal, predator-prey and bouncing ball with friction, our proposed framework obtains a higher $R^2$ score (a score of 1 corresponds to perfect regression) implying that the learned dynamics modulators have captured information of the underlying parameters of the system.
>
> Static modulators: intuitively they capture static characteristic features of the data, in the case when the data is image sequences, for example. In this case, the static modulator would capture the color, shape and other static features of the moving object.
>
> To assess whether this assumption is true we evaluate the learned vector space of the static modulator for the rotating MNIST experiment. In particular, in Figure 16., Supplementary material D, Additional results, we see that, even though we have not explicitly enforced any topology in the latent vector space, we can observe a clustering of the data points corresponding to a given digit style (0 versus 1 for example). This qualitative observation indicates that the static modulator has captured information of the underlying characteristic of the observations. Albeit the separation of the clusters is not perfect, the explanation of this is two-fold: visually examining the data points it can be seen that some digits from different classes look very similar due to the handwriting, and we have not enforced in the latent space that each digit class should be embedded as far as possible from each other, hence the observed overlap is reasonable.
>
> We will provide a sufficient visual description in the final version of the paper.

---

> > ### Comment · Reviewer_Y7gY · 2023-08-12
> >
> > Thank you very much for the reply.  I will leave the score as is since my original concerns are not fully addressed.

---

> > > ### Author Response · Authors · 2023-08-12
> > >
> > > Thank you for responding. Could you please clarify what part of your concern is not addressed so we can respond appropriately? As we have showcased that (1) the obtained performance is not due to the increased parameter count; (2) MoNODE outperforms NODE also on chaotic systems; (3) provided both visual and intuitive explanation of the modulator variables.

---

### Official Review · Reviewer_m2DA · 2023-07-11

**Soundness:** 3 good
**Presentation:** 3 good
**Contribution:** 3 good
**Rating:** 6
**Confidence:** 3

**Summary:**

The paper proposes an approach for generalizing NeuralODE to different system parameters, in the setting of a single dynamical system with different parameters. A dynamic and static modulators are applied to specific parts of the encode-process-decode architecture: the dynamic modulator acts on the NeuralODE part, while the static modulator only acts on the decoder. They are learned in an end-to-end manner.
Evaluation is performed on synthetic datasets, as well as a real dataset (walking humans), by using different NODE variants. It showcases the generalization capabilities of the 2 modulators.
The paper’s idea is interesting, and I found the experiments are convincing.

**Strengths:**

- The proposed framework is easy-to-integrate to existing NODE methods to increase their performance.
- The latent space is more interpretable than with vanilla NODE, and provides high correlation wrt the dynamics parameters
- The proposed method drastically improves the used NODE methods for long-range prediction

**Weaknesses:**

-

**Questions:**

- Is there a way the modulations spaces could be used for interpolation between different systems?
- Could this work be extended to process PDE equations?

**Limitations:**

The work only studies dynamical systems up to 3 dimensions, would it be able to scale to higher dimensions?

---

> ### Author Rebuttal · Authors · 2023-08-08
>
> Thank you for the review, please find below our response for the questions voiced.
>
> > Is there a way the modulations spaces could be used for interpolation between different systems?
>
> Thank you for the interesting question. It touches upon a possible Bayesian extension of the presented work. In particular, in the current set-up the modulator variables are point estimates, however, this could be adjusted to learning possibly multi-modal modulator variable distributions. Ideally, each mode of the inferred posterior could correspond to a particular system (e.g., a handful of particles with no/weak/strong interactions). That being said, the completely black-box nature of our framework could make it difficult to interpret/identify modulators and therefore the systems.
>
> > Could this work be extended to process PDE equations?
>
> Based on the recent work on neural PDEs [1], it is possible to parameterize the systems of ODEs resulting from spatial discretization of a PDE in the same way as our framework. An interesting research question would be, for instance, learning modulators as a function of spatial coordinates.
>
> [1] Learning Space-Time Continuous Neural PDEs from
> Partially Observed States. Iakovlev, Heinonen, Lähdesmäki, 2023.

---

> > ### Comment · Reviewer_m2DA · 2023-08-19
> >
> > Thank you for answering my questions, I will keep my rating

---

### Official Review · Reviewer_VxYT · 2023-07-12

**Soundness:** 3 good
**Presentation:** 3 good
**Contribution:** 2 fair
**Rating:** 6
**Confidence:** 3

**Summary:**

The authors introduce Modulated Neural ODEs by incorporating a static modulator that averages observation embeddings across the whole time and a dynamics modulator that averages observation embeddings by time subsequences. The proposed model consistently improves the existing model's ability to generalize to new dynamic parameterizations.

**Strengths:**

The presentation is pretty straightforward and the idea is simple yet effective, as shown by experiments.

**Weaknesses:**

- As the output of the static modulator and the dynamics modulator is used by concatenation, I would like to see the experimental results without the static modulator or the dynamics modulator to determine if they are both essential.
- I would like the authors to add the inference time cost comparison with and without those modulators.

**Questions:**

See weaknesses.

**Limitations:**

The authors have already discussed the limitations of this work, e.g., the current exploration is only limited to deterministic systems and the performance of the work is largely affected by the base model’s performance.

---

> ### Author Rebuttal · Authors · 2023-08-08
>
> Thank you for reviewing our work, please find below the response to the concerns voiced.
>
> > As the output of the static modulator and the dynamics modulator is used by concatenation ...
>
> We never concatenate static and dynamic modulators. They are additional inputs to the decoder and differential function, respectively. We do concatenate the dynamic state $z$ and each modulator variable separately.
>
> > ... I would like to see the experimental results without the static modulator or the dynamics modulator to determine if they are both essential.
>
> We do not always use both modulator variables, but rather the one(s) that match the problem:
>
> (i) for low-dimensional problems with changing dynamic's parametarization we use **only** dynamics modulator (Section 5.1)
>
> (ii) for high-dimensional problems with changing object features, but fixed dynamics, we use **only** static modulator (Section 5.2)
>
> (iii) for problems that have both, changing dynamic's parametrization and changing object features, we use **both** modulator variables (Section 5.3)
>
> That said, we investigated the utility of using both modulator variables versus only one for Mocap (Section 5.3). Using both modulator variables resulted in the lowest test MSE as compared to using only one, either static or dynamics modulator, as reported in Table 12 (Supplementary material D, Section 'Additional results'). This performance gap can be attributed to the data, which contains both static (length of limbs) and dynamics (length of legs) modulating signals, therefore, both variables are needed for optimal performance.
>
> If for problems (i) and (ii) the corresponding modulator is switched to the opposite ((i) Sinusoidal data with static modulator instead of dynamics; and (ii) rot. MNIST with dynamics modulator instead of static) we see a drop in performance with respect to test MSE. Specifically, for Sinusoidal data for MoNODE + static mod. the performance drops to 2.68 (0.38) from MoNODE + dyn. mod. 0.29 (0.11). For Rot. MNIST MoNODE + dyn. mod. the performance drops to 0.096 (0.008) from MoNODE + static mod. 0.030 (0.001). In all, this confirms that the modulator variable selection is essential, depends on the problem at hand and our framework can accommodate multitude of problem set-ups.
>
> > I would like the authors to add the inference time cost comparison with and without those modulators.
>
> We compute the inference time cost for the Sinusoidal data experiment, where the test data consists of 50 trajectories of length 150. We record the time it takes NODE and MoNODE to predict future states while conditioned on the initial 10 time points. We perform the evaluation 10 times and report the mean (in seconds) and std. The inference time cost for NODE is 0.312 (0.050), while for MoNODE is 0.291 (0.040). We are faster because the number of parameters for MoNODE is lower than for NODE, 24598 versus 24666, respectively. As such, we would like to highlight that the inference time costs of our framework is on par with the base models where the exact time is dependent on the total number of parameters, which is comparable between all base and Mo*NODE models (Table 7., Supp. mat. C). We will add these results to the final version.

---

> > ### Comment · Reviewer_VxYT · 2023-08-13
> > **Response to Authors**
> >
> > Thank you for solving my questions. I will raise my score.

---

### Official Review · Reviewer_8J7y · 2023-07-14

**Soundness:** 3 good
**Presentation:** 3 good
**Contribution:** 2 fair
**Rating:** 6
**Confidence:** 5

**Summary:**

Existing neural ordinary differential equations (NODEs) present the problem of variation across trajectories only through initial state values. As a solution to the problem, Modulated Neural ODE (MoNODE), which improves the existing NODE method by distinguishing dynamic and static states, is presented. In MoNODE, 1) static modulators and 2) dynamics modulators are used to divide dynamic variables from time-invariant variables. By attaching these two modulators to existing NODE studies, they show good performance in various datasets.

**Strengths:**

1. They presented the problem of handling static variables well in NODE, and suggested dynamics modulator and static modulator as a solution. This is very intuitive.

2. It is very attractive that it can be applied to any NODE model and does not have to be divided into static and dynamic variables in advance.

3. They showed the performance using various datasets to evaluate dynamic and static modulators.

**Weaknesses:**

1. They experimented by dividing 5.1 dynamics modulator variable and 5.2 static modulator variable data to evaluate their modulators, but they could not show whether each modulator actually played its role.


**Questions:**

In experiments 5.1 and 5.2, static variable and dynamics variable data were separately tested to evaluate each static modulator and dynamics modulator. Was the dynamics modulator also applied to static variables in these experiments? If the dynamics modulator was used even in the static variable experiment, you should remove the dynamics modulator.

If you did not use dynamics modulator in static variable experiment, you should experiment using dynamics modulator. In other words, through these experiments, it is necessary to check whether each modulator is playing a role.


**Limitations:**

There is no limitations

---

> ### Author Rebuttal · Authors · 2023-08-08
>
> Thank you for the review, please find below our response to the concerns voiced.
> > Was the dynamics modulator also applied to static variables in these experiments? If the dynamics modulator was used even in the static variable experiment, you should remove the dynamics modulator. If you did not use dynamics modulator in static variable experiment, you should experiment using dynamics modulator. In other words, through these experiments, it is necessary to check whether each modulator is playing a role.
>
> To confirm the role of each modulator variable we have performed 2 additional ablations with the MoNODE framework on: (a) Sinusoidal data with static modulator instead of dynamics; and (b) rot. MNIST with dynamics modulator instead of static.  We report the test MSE across 3 different initialization runs with standard deviation. For Sinusoidal data for MoNODE + static mod. the test MSE performance drops to 2.68 (0.38) from MoNODE + dyn. mod. 0.29 (0.11). For Rot. MNIST MoNODE + dyn. mod. the performance drops to 0.096 (0.008) from MoNODE + static mod. 0.030 (0.001). In addition, we examined the latent embeddings of the dynamics modulator for Rot. MNIST. Where previously for the content modulator we observed clusters corresponding to a digit's class (Fig. 16, Supp. mat. D), for dynamics modulator such a topology in the latent space is not present, see Fig. 2 (additional PDF, general response). In all, the results confirm that the modulator variables are correlated with the true underlying factors of variation as indicated by the obtained $R^2$ scores (Table 3., main paper) and latent embedding plots (Fig. 16, Supp. mat. D), and, as such, play their corresponding roles.

---

> > ### Comment · Reviewer_8J7y · 2023-08-10
> > **Response to Authors**
> >
> > Thank you for solving my concerns. Even considering other reviewers' reviews, I'll raise my score.

---

### Official Review · Reviewer_dvMp · 2023-07-14

**Soundness:** 2 fair
**Presentation:** 4 excellent
**Contribution:** 1 poor
**Rating:** 4
**Confidence:** 4

**Summary:**

The paper propose to improve long term forecasting abilities of NODEs by incorporating the dynamics with learnt static parameters. The system is composed of a static modulator and a dynamic modulator, the final solution is sampled from a distribution conditioned on the static modulator and the dynamic generated using the dynamic modulator. Empirical results suggests improvements over NODE on time simple series datasets.

**Strengths:**

Presentation is clear, and the paper is well written.

The empirical improvements on time series data is significant on the long run.

**Weaknesses:**

Similar work has been proposed, see https://arxiv.org/abs/2302.13262 for an example. Although the methods are slightly different, the fundamental idea of incorporating with static parameters are the same.

There are no clear theoretical guarantees that the method will work better.

The network structure is very complicated, the reviewer is worried about the training difficulties. It would be helpful to see some ablation and show it is easy to train.

**Questions:**

How are the networks trained? There are so many different networks. Also, how is this method significantly different from existing literature?

---

> ### Author Rebuttal · Authors · 2023-08-08
>
> Thank you for reviewing our work. Please find below our response to the concerns voiced.
> > How are the networks trained? There are so many different networks.
>
> We have a simple feedforward architecture,  which we can train easily end-to-end, no more complex than NODE [1] or ODE2VAE [2]. Specifically, MoNODE optimizes ELBO like NODE, that is the optimization of the additional modulator encoder networks is implicit, with no extra loss terms:
>
> $\textup{log} \ p(X)  \geq \textup{ELBO} = \sum_n E_{q_{\nu}} \left[  \textup{log} \ p\big(x_{1:T}^{n} \mid z_1^{n} \big)  \right] - \textup{KL}\left[ q_{\nu}( z_1^{n}\mid x^n_{1:T_z}) || p(z_1) \right].$
>
> We would like to clarify that there are only 3 networks in the proposed method: two of which are the standard building blocks of a VAE (encoding, decoding network) as in [1] and the third, our addition, the modulator prediction network. We explicitly name the networks to clarify the intuition, following the narrative style of ODE2VAE [2].
>
> [1] Chen, R. T., Rubanova, Y., Bettencourt, J., \& Duvenaud, D. K. (2018). Neural ordinary differential equations. Advances in neural information processing systems, 31.
>
> [2] Yildiz, C., Heinonen, M., \& Lahdesmaki, H. (2019). ODE2VAE: Deep generative second order ODEs with Bayesian neural networks. Advances in Neural Information Processing Systems, 32.
>
>  > It would be helpful to see some ablation and show it is easy to train.
>
> The proposed model is very easy to train, and not sensitive to hyperparameters. Our model performs consistently well under different hyperparameterization:
>
> (i) the solver used (Table 9., Supp. mat. D);
>
> (ii) the number of input frames $T_z$ used (Table 10., Supp. mat. D);
>
> (iii) the latent dimensionality of the latent ode state $q_z$ and modulator variable $q_s$ (Table 11., Supp. mat. D).
>
> In addition, to showcase that our proposed framework is easy to train we have plotted the validation MSE versus wall clock time during training for sinusoidal data, please see Figure 1 (additional PDF, global response). As it as apparent from the figure, our framework is easier to train than all baseline methods. We realize that these ablations were not easily accessible, so we will move a highlight of them to the main paper.
>
> > no clear theoretical guarantees
>
> Similarly to earlier works in this area of research like ODE2VAE [2], LatSegODE [3] and Neural Event ODE [4], we introduce an additional structure on top of the flexible neural ODE model class without a theoretical investigation. Nonetheless, we show empirically that in practice our framework brings a significant accuracy improvement with an average of 55.25\% across all experiments, and a high of 88.38\% improvement on PP dataset. We leave a theoretical study as a fantastic next step for future work, to explore the limits of our ideas, especially with respect to the complexity of the static signal.
>
> [3] Shi, R., & Morris, Q. (2021, July). Segmenting hybrid trajectories using latent odes. In International Conference on Machine Learning (pp. 9569-9579). PMLR.
>
> [4] Chen, R. T., Amos, B., & Nickel, M. (2020). Learning neural event functions for ordinary differential equations. arXiv preprint arXiv:2011.03902.
>
> > Also, how is this method significantly different from existing literature?
>
> We compare with a great array of state-of-the-art methods, a visual summary is in Table 1 in the main paper. Our method is significantly different from the existing methods as
>
> (i) we introduce modulator variables that allow the model to preserve core dynamics while being adaptive to modulating factors. This, in turn, allows us to model underlying factors of variation that are fixed in time.
>
> (ii) our framework addresses a common limitation of existing NODE type models: poor long-term forecasting performance.
>
> (iii) our modulator variables are informative of the true unknown factors of variation as measured by $R^2$ scores.
>
> (iv) it is a *general* framework that can be combined with a large variety of existing NODE type models, bringing on average across all experiments 55.25\% accuracy improvements.
>
> > similar work has been proposed, see https://arxiv.org/abs/2302.13262
>
> We do appreciate your knowledge of the related work, even of its newest additions. As you might have noticed, this arxiv submission is very related to ours and we are very aware of it. On the other hand, the paper isn't published yet, and we have very good reasons not to cite it.

---

> > ### Comment · Reviewer_dvMp · 2023-08-11
> >
> > Thank you for your response. I will leave my score as is.

---

> > > ### Author Response · Authors · 2023-08-12
> > >
> > > Thank you for your fast response. We wanted to point out that we addressed in a very direct way at least two of three of your concerns (similarity with related work and ease of training), and agreed with the third, the lack of theoretical guarantees, but also pointed out that it is standard in the field. We are a bit confused then, why even after addressing them, the qualitative evaluation did not change and would be happy to discuss further.

---

> > > > ### Comment · Reviewer_dvMp · 2023-08-13
> > > >
> > > > The response is helpful for my understanding. However, the contributions of this work are not sufficient for this conference.

---

### Author Rebuttal · Authors · 2023-08-09

We sincerely appreciate your time and effort in providing valuable feedback. Since our findings and core ideas haven't resonated with you as much as they do with us, we'd like to take this opportunity to first try to better convince you of the overall approach and significance of our work. For specific comments, we have addressed them in the respective "rebuttal" sections.

**Why modulators:**
Standard NODEs and subsequent works aim to learn a single dynamics function that has a single latent ODE state.  In practice, however, different observations might have different fixed underlying parameterization of the dynamics function or different observed characteristics. Whereas standard methods can only model this variation through a **changing** latent ODE state resulting in a complicated dynamics function that fails to extrapolate (see Fig 2-4, Table 3), Modulated NODEs explicitly accounts for this variation through the time-invariant modulator variables.

**Benchmarking:**
We assessed our method on five established NODE benchmarks, enhancing their intricacy by sampling all parameters, incorporating friction, utilizing all digits, adding more walking sequences, and so on. It was striking to observe how easily prior methods, including the state-of-the-art HBNODE, collapsed due to these minor dataset modifications.
We believe that this highlights a general limitation of earlier methods and affirms the importance of our approach.


**Findings:**
Our method on average improves long-term forecasting test MSE by 55.23\% across all baselines and datasets. Moreover, the modulator variables' latent representation offers insights into the true unknown factors of variation: (i) they align closely with the ground truth factors as measured by $R^2$ (Table 3), (ii) they exhibit a similar topology to the observation space (Figure 16, Supp. mat. D). As a consequence, also the latent ODE state is more consistent with the true underlying dynamics: (i) the amplitudes of latent trajectories heavily correlate with the observed sequences (Figures 2-3), (ii) the rotating digit sequences overlap as expected (Fig 4).
In conclusion, our method demonstrably excels at identifying meaningful representations and offers an elegant approach that opens new avenues in dynamic representation learning, given its wide applicability.

**Applicability:**
Our intent was to propose a "simple-yet-effective" solution, enabling easy integration into various NODE models. The empirical results on four NODE variants attest to the immediate advantages our modulator variables offer. We would be pleased if you agree that the simplicity and versatility of our method warrants positive recognition.

---

### Decision · Program_Chairs · 2023-09-21

**Decision:**

Accept (poster)

**Comment:**

Six experts reviewed this paper with mixed scores. The area chairs agree that this work makes a very important contribution by introducing new Modulated Neural ODEs. The reviewers did raise some valuable concerns that should be addressed in the final camera-ready version of the paper